# Fine-mapping and molecular characterisation of primary sclerosing cholangitis genetic risk loci

Elizabeth C. Goode [1,2,3], Laura Fachal [1], Nikolaos Panousis[1], Loukas Moutsianas [1], Rebecca E. McIntyre[1], Benjamin Yu Hang Bai [1,2], Norihito Kawasaki[4], Alexandra Wittmann[4], Tim Raine[2], Simon M. Rushbrook[3,5] & Carl A. Anderson [1] ✉

Genome-wide association studies of primary sclerosing cholangitis have identified 23 susceptibility loci. The majority of these loci reside in non-coding regions of the genome and are thought to exert their effect by perturbing the regulation of nearby genes. Here, we aim to identify these genes to improve the biological understanding of primary sclerosing cholangitis, and nominate potential drug targets. We first build an eQTL map for six primary sclerosing cholangitis-relevant T-cell subsets obtained from the peripheral blood of primary sclerosing cholangitis and ulcerative colitis patients. These maps identify 10,459 unique eGenes, 87% of which are shared across all six primary sclerosing cholangitis T-cell types. We then search for colocalisations between primary sclerosing cholangitis loci and eQTLs and undertake Bayesian fine-mapping to identify disease-causing variants. In this work, colocalisation analyses nominate likely primary sclerosing cholangitis effector genes and biological mechanisms at five non-coding (UBASH3A, PRKD2, ETS2 and AP003774.1/CCDC88B) and one coding (SH2B3) primary sclerosing cholangitis loci. Through fine-mapping we identify likely causal variants for a third of all primary sclerosing cholangitis-associated loci, including two to single variant resolution.

Primary Sclerosing Cholangitis (PSC) is a rare immune-mediated inflammatory disease of the bile ducts that affects one in 10,000 people in developed countries. PSC confers risk of serious disease sequelae including hepatobiliary malignancy and progression to end-stage liver failure[1]. Inflammatory bowel disease (IBD), and predominantly ulcerative colitis (UC), is present in up to 80% of patients with PSC[2,3]. Our current limited understanding of the aetiology and pathogenesis of PSC underpins the absence of effective medical therapies that halt or attenuate disease progression. Thus, liver transplantation is currently the only treatment option for those with advanced disease and PSC remains the fifth most common indication for liver transplantation in Europe[4].

Genome-wide association studies (GWAS) have been widely applied across common complex diseases to extract insights into disease pathogenesis. Twenty-three regions of the human genome have been robustly associated with PSC susceptibility via GWAS[5–7]. However, the majority of these association signals reside outside of genes, in the non-coding regions of the genome, making it difficult to causally implicate particular genes and biological mechanisms in PSC risk. Furthermore, the causal variant underpinning the association signal is typically unknown, hindering efforts to undertake downstream functional experiments to understand the biological consequences of these variants. Given that drug mechanisms with genetic support are twice as likely to succeed from phase I trials to approval

[1]Wellcome Sanger Institute, Hinxton, Cambridge, UK. [2]University of Cambridge, Cambridge, UK. [3]Norfolk and Norwich University Hospital, Norwich, UK. [4]Quadrum Institute, Norwich, UK. [5]Norwich Medical School, University of East Anglia, Norwich, UK. ✉e-mail: carl.anderson@sanger.ac.uk

than those without[8], it is important that research is undertaken to identify PSC effector genes at disease-associated loci[9].

In IBD, as with other immune-mediated diseases[10], identifying disease association signals that share causal variants with genetic loci associated with variation of expression of a nearby gene (cis-eQTLs) has proven to be a successful means of connecting non-coding disease association signals with the genes they dysregulate. Furthermore, we have recently shown that the disease-causing variant underpinning an association signal shared with an eQTL can often be more readily identified via fine-mapping the genetic effect on gene expression, rather than by fine-mapping the disease association itself[11].

Genetic effects on gene expression are known to vary between tissues and environments[12,13]. As such, eQTL maps in disease-relevant cell-types and states must be available if these so-called 'colocalisation' analyses are to identify disease-relevant effector genes and causal variants[14,15]. Genetic effects on gene expression have been well characterised for many immune cells in the blood[13,16,17], creating a rich resource for our colocalisation analyses across PSC-associated risk loci. However, genetic effects on gene expression in key PSC cell types such as gut-homing T-cell subsets remain uncharacterised. Two hypotheses of the causal pathogenesis of PSC involve T-cells. Either via the adaptive immune system, but also through the dual-homing (gut-liver) homing T-cell subtypes (CCR9 + ) hypothesis[18]. However, no studies have yet mapped eQTLs in cells ascertained from patients with PSC, which may be necessary to understand the eQTL landscape of PSC.

In this study, we present a genetic map of gene expression regulation (eQTL map) for six PSC-relevant T cell subsets ascertained from PSC and UC patients. To identify effector genes and thus potential drug targets at PSC risk loci we search for shared causal variants (colocalisations) between eQTLs and PSC loci, enriching these analyses with additional molecular QTLs maps available in the public domain, across a wide range of PSC relevant tissues and cell types. To pinpoint PSC causal variants, we undertake fine-mapping of disease-associated loci and any colocalising molecular QTLs.

## Results

### PSC variants are significantly enriched at T-cell regulatory elements

To identify cell types likely to be playing a role in PSC pathogenesis we ran S-LDSC analysis across two large independent datasets, the immune cell atlas[19] (Supplementary Fig. 1, top panel) and the cis-elements atlas of human tissues[20] (Supplementary Fig. 1, bottom panel). Within the cis-elements atlas, the most significant cell types were both T-cell populations (Lymphocyte 1 (CD8 + ) and T lymphocyte 2 (CD4 + )), although none of the tests passed multiple testing correction. Within the immune cell atlas (Supplementary Fig. 1 top panel), the only significantly enriched cell types were T-cells, both stimulated (CD8 + ; Central and Effector memory CD8 + ; Effector CD4 + ; Follicular T helper; Gamma delta; Memory and Naive T effector; Regulatory; Th1, Th2 and Th17 precursors) and unstimulated (Th17 precursors) from the immune cell atlas. These results support our decision to focus our de novo eQTL mapping on understudied T cell subsets of potential relevance to PSC.

### PSC-specific T-cell eQTL data

We mapped cis-eQTLs for six T-cell subtypes, derived from the peripheral blood of 76 individuals with PSC-UC or UC. Cohort characteristics are shown in Supplementary Table 1. After extracting all significant eQTL/gene pairs, we detected a median of 1337 eQTLs per cell type (5% FDR), with >90% of significant eQTLs falling within 1 Mb of a transcription start site (Supplementary Fig. 2). Differential gene expression analyses failed to find any differences in gene expression between the PSC-UC and UC samples for each of the six T-cell subsets, a finding is not entirely unexpected given that both groups share the

UC phenotype. Thus, we then undertook eQTL mapping in the PSC-UC and UC samples together, to maximise power. Using *mashR* to model the sharing of eQTLs across the six cell-types, we mapped significant eQTLs for 10,459 unique genes (eGene - a gene whose expression level is associated with a genetic variant at 5% lfsr), a number more than three times the sum of all significant, unique eGenes detected in the individual cell-type analyses. Of these 10,459 unique eGenes, 87% (9176) were shared across all six cell types, while 4.7 % (489) were specific to a single cell type (Supplementary Fig. 3). Gene Ontology (GO) analysis of the eGenes using *g:profiler*[21] did not highlight any gene sets or pathways enriched with cell-type specific or shared eQTLs.

On average, 61–69% of the significant eGenes identified in PSC T cells replicate in GTEx tissues (estimated as the average pi1 per PSC T cell population across all GTEx tissues). The GTEx tissue with the highest replication rates was whole blood (ranging from an 88% replication rate of PSC treg eGenes to a 93% replication rate of PSC CD4posCCR9pos eGenes) (Supplementary Fig. 4). The average replication rates in eQTL datasets composed of immune cell types were, as expected, larger. For example, the average replication rate per PSC T cell type in Schmiedel et al.[22] (which includes 15 immune cell types under resting and stimulated conditions) ranged between 75–82%; with the larger replication rates on this study observed with Th17 cell and Treg naive. In the blueprint data[23] (which includes monocytes, neutrophils and CD4 + T cells) average replication rates ranged between 90–92%, with the larger replication rates observed in CD4 + T cells.

### Colocalisation with molecular QTLs

We conducted colocalisation between PSC loci and 48 molecular QTL datasets covering five gastrointestinal whole-tissue types, eleven immune cell-types (including our six PSC-specific T-cells) and five different molecular QTL types (Supplementary Data 1). We identified colocalisations with one or more eQTLs for four of the fifteen PSC loci (Supplementary Data 2). Of these, three colocalised with a single gene in the associated locus, and one (Chr11:64107735) colocalised with two eGenes; *APO03774.1* in PSC-T-cells and *CCDC88B* in monocytes. All four of these loci also colocalised with another molecular QTL; two with methQTLs, one with a histQTL and one with a spliceQTL. For three of the four colocalising loci, Chr19:47205707 (*PRKD2*), Chr21:40466744 (*ETS2*) and Chr21:43855067 (*UBASH3A*), the same eGene colocalised with the PSC association signal in more than one cell-type or tissue, further increasing our confidence in a pathogenic role for these genes. Regarding our PSC-specific T-cell subtypes, two of the four colocalising PSC loci colocalised with eQTLs in multiple PSC T-cell subtypes; Chr21:43855067 colocalised with an eQTL for *UBASH3A* in five of the six PSC specific T-cell subtypes, and Chr11:64107735 colocalised with an eQTL for *APO03774.1* in four of the six PSC specific T-cell subtypes (Supplementary Data 2).

Using our own PSC T-cell eQTL maps, we also performed colocalisation of PSC T-cell eQTLs with UC, CD, rheumatoid arthritis (RhA) and Type 1 diabetes mellitus (T1DM) risk loci, identifying ten IMD risk loci that colocalised with eQTLs for one or more genes (Supplementary Table 2). Notably, the Chr1 rs3180018 UC risk locus colocalised with an eQTL for *GBAP1* in T-reg, T-memory, CD4 + CCR9- and CD4 + CCR9 + T-cells (PP4 = 0.91). The UC Chr7: 128573967 risk locus colocalised with an eQTL for *IRF5* in T-memory cells.

### Fine-mapping of PSC loci in PSC and molecular QTL data

Fine-mapping of the PSC associated loci identified nineteen independent association signals across the fifteen risk loci (Fig. 1). Credible sets ranged in size from one to sixty-two variants (Table 1). Review of the 1000 Genomes Project and UK10K data-sets identified that across all loci there were no missing SNPs in high LD (r² > 0.8) with the most probable causal variants, giving us confidence in the output of the fine-

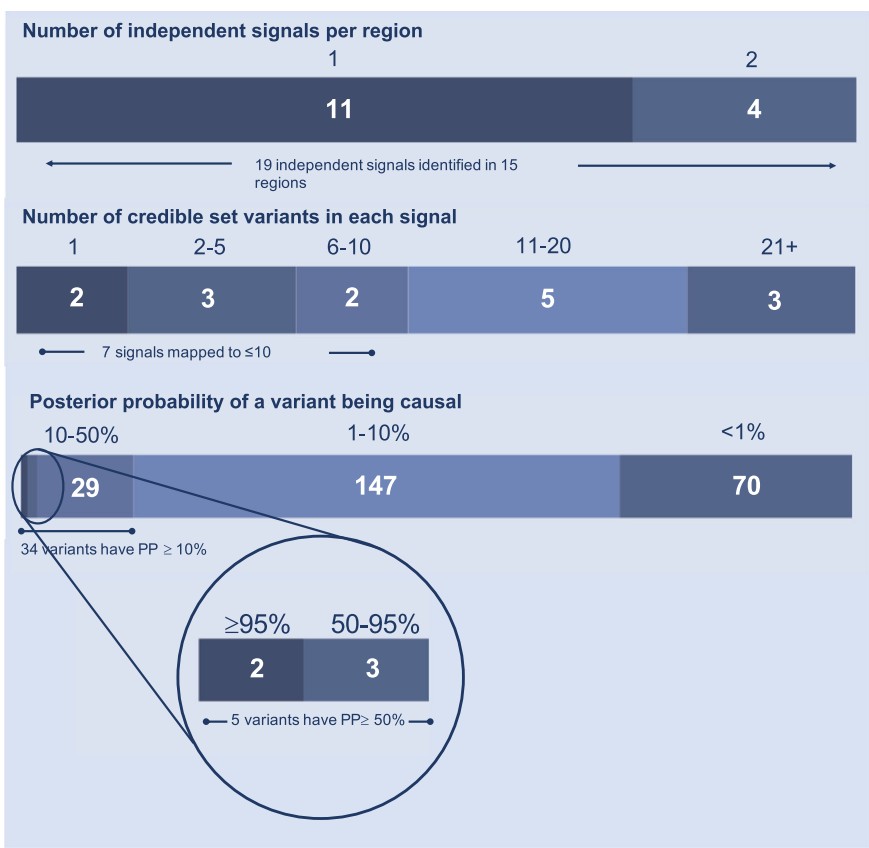

**Fig. 1 | Summary of fine-mapping.** Summary of the fine-mapping results across 15 PSC risk loci.

## Table 1 | Fine-mapping of PSC risk loci

| Region | | | | | Fine-mapping | | | |
|---|---|---|---|---|---|---|---|---|
| Chr | Candidate gene | Lead GWAS SNP | Position (b37) | Signal | SNP PP$_{max}$ | Position (b37) | PP Causal | Credible set size |
| 1 | MMEL1 | rs3748816 | 2526745 | 1 | rs61763697 | 2810791 | 0.07 | 62 |
| 2 | BCL2l1 | rs72837826 | 111933001 | 1 | rs72837826 | 111933001 | 0.18 | 12 |
| 2 | CD28 | rs7426056 | 204612058 | 1 | rs5837875 | 204647878 | 0.19 | 6 |
| | | | | 2 | rs231799 | 204707417 | 0.17 | 24 |
| 3 | MST1 | rs3197999 | 49721532 | 1 | rs11716895 | 49762779 | 0.11 | 13 |
| | | | | 2 | rs13083791 | 49721798 | 0.07 | 122 |
| 3 | FOXP1 | rs80060485 | 71153890 | 1 | rs80060485 | 71153890 | **0.99** | 1 |
| | | | | 2 | rs36023390 | 71523093 | 0.14 | 2 |
| 4 | IL2-IL21 | rs13140464 | 123499745 | 1 | rs13119723 | 123218313 | 0.09 | 50 |
| 6 | BACH2 | rs56258221 | 91030441 | 1 | rs7750271 | 91036225 | 0.20 | 12 |
| 10 | IL2RA | rs4147359 | 6108439 | 1 | rs4147359 | 6108439 | 0.46 | 5 |
| 11 | CCDC88B | rs663743 | 64107735 | 1 | rs35247680 | 63884747 | **0.61** | 2 |
| | | | | 2 | rs663743 | 64107735 | 0.41 | 2 |
| 12 | SH2B3 | rs3184504 | 111884608 | 1 | rs3184504 | 111884608 | **0.99** | 1 |
| 16 | CLEC16A | rs725613 | 11169683 | 1 | rs725613 | 11169683 | 0.16 | 12 |
| 18 | CD226 | rs1788097 | 67543688 | 1 | rs1610555 | 67543147 | 0.08 | 44 |
| 19 | PRKD2 | rs313839 | 47221557 | 1 | rs313839 | 47221557 | 0.23 | 14 |
| 21 | ETS2 | rs2836883 | 40466744 | 1 | rs4817988 | 40468838 | **0.58** | 10 |
| 21 | UBASH3A | rs1893592 | 43855067 | 1 | rs1893592 | 43855067 | **0.62** | 5 |

Fine-mapped loci (PP of casuality ≥0.5) shown in bold.

mapping algorithm. We resolved two loci to a single causal variant with >95% certainty and three loci to larger credible sets where one variant was assigned >50% posterior probability of causality (Fig. 1). For loci colocalising with a molecular QTL, fine-mapping of the colocalising molecular QTL data improved our fine-mapping resolution for two loci; the credible causal set for Chr19:47205707 (*PRKD2*) reduced from fourteen to eight variants, and the credible causal set for Chr21:438655067 (*UBASH3A*) reduced from five to a single causal variant (Table 2). Notable fine-mapping and colocalisation results are further discussed on a per-locus basis below.

**Table 2 | Fine-mapping of PSC rick loci in functional QTL data**

| Chr | GWAS Fine-mapping | | | Colocalisation | | | MolQTL Fine-mapping | | |
|---|---|---|---|---|---|---|---|---|---|
| | GWAS Fine-map SNP | Posterior Probability of causality | Credible set size | QTL type | Cell type | Gene | MolQTL SNP PP$_{max}$ | PP$_{max}$ | Credible set size |
| 11 | rs663743 | 0.41 | 2 | eQTL | Monocyte | *CCDC88B* | rs663743 | 0.03 | 245 |
| 19 | rs313839 | 0.23 | 14 | eQTL | Monocyte | *PRKD2* | rs112445263 | 0.14 | 8 |
| 21 | rs4817988 | 0.58 | 10 | eQTL | Monocyte | *ETS2* | rs4817987 | 0.07 | 47 |
| | | | | H3K27AC | Monocyte | N/A | rs2836878 | 0.13 | 11 |
| 21 | rs1893592 | 0.61 | 5 | eQTL | CD4 + T-cell | *UBASH3A* | rs1893592 | 1.00 | 1 |
| | | | | SpliceQTL | CD4 + T-cell | *UBASH3A* | rs1893592 | 1.00 | 1 |

## The AP003774.1/CCDC88B locus

We identified colocalisations between the Chr11:64107735 rs663743 PSC risk locus and an eQTL for *AP003774.1* in five cell types with ≥95% PP4; T-regulatory, T-memory and CD4 + CCR9- effector-memory T-cells and CD8 + CCR9- effector-memory T-cells from patients with PSC-UC and lone-UC, EBV-transformed lymphocytes and whole blood from healthy volunteers. There was some additional evidence (PP4 72%) to support colocalisation with the same gene in CD4 + CCR9 + T-cells. The PSC risk increasing rs663743*G allele was associated with reduced expression of *AP003774.1* across all colocalising cell-types. This PSC locus also colocalised with an eQTL for a second gene, *CCDC88B* (PP4 0.85) in monocytes (Supplementary Fig. 5). We used ENCODE data to demonstrate that this locus overlaps H3K27me3, a marker of an inactive or silenced regulatory region, in keeping with our finding that the PSC risk increasing allele reduces expression of *AP003774.1*. We fine-mapped this locus within the PSC data, to a single variant, rs663743 at Chr11:64107735 (single variant credible set).

This pleiotropic locus is also associated with T1DM disease susceptibility (PP4 PSC-T1DM 82%). A recent T1DM fine-mapping effort identified one disease susceptibility signal at this locus, with rs663743 being the variant most likely to be driving the association (out of a large credible set with 72 variants[24]). Fine-mapping this locus with the PSC GWAS data identified two independent disease susceptibility signals (Table 1), with each credible set comprising only two variants. The most likely variants driving these two independent signals are rs663743 (PP 41%) and rs35247680 (PP 61%).

## The ETS2 locus

The Chr21:40466744 locus colocalised with an eQTL for *ETS2* in whole blood, monocytes and IL-4 stimulated macrophages (Fig. 2). Furthermore, it colocalised with an H3K27ac histQTL (a histone acetylation marker associated with higher activation of transcription) in both monocytes and neutrophils, suggesting increased expression of *ETS2* may be mediated by increased transcription factor binding affinity. The PSC *ETS2* locus also colocalised with an association signal for neutrophil counts (PP4 83%), where the PSC risk increasing allele was associated with a reduction in neutrophil count.

We fine-mapped the *ETS2* locus in the PSC data to a credible set containing ten variants in high LD ($r^2 > 0.8$). The most probable causal variant was rs4817988 (PP 58%), followed by rs2836884 (8% PP) and rs2836883 (5% PP). Although rs4817988 is located in a non-coding region of the genome, it overlaps a CTCF transcription factor binding site. This association signal is also shared with CD and UC (PP4 PSC-CD, PP4 PSC-UC > 80%) (Table 3). It has also been the subject of two IBD fine-mapping studies, in which it was resolved to a ten variant 95% credible set[25]; and a seven variant 99% credible set[24]. This last study also undertook functional follow up of the credible set, identifying rs2836882 as the most likely variant driving the expression of *ETS2*. Eight of these fine mapped variants in previous studies overlapped with the PSC credible set (including rs4817988), with 39% of the PP attributed to rs9977672 (PP = 0.01 for this variant in the PSC data).

## The PRKD2 locus

The Chr19:47205707 locus colocalised with an eQTL for *PRKD2* in monocytes (PP4 = 94%) (Fig. 3). The PSC risk increasing allele was associated with decreased expression of *PRKD2*. This locus also colocalised with genetic variants regulating two CpG methylation sites (cg00838415 and cg08634012) in both monocytes and neutrophils, suggesting that repression of *PRKD2* expression may occur via hypermethylation. Although this locus also colocalised with an eQTL for *PRKD2* in transverse and sigmoid colon tissue (PP4 94%) and this locus is also associated with IBD risk, in keeping with our previous study[7] our analysis supported a different causal variant driving the IBD and PSC associations in this region (PP3 for colocalisation of the PSC locus with UC and CD, 94% and 97% respectively). However, this PSC locus did colocalize with a T1DM risk locus, which has been previously reported as an eQTL for PRKD2 in monocytes[15]. Thus, our results suggest that lower expression of PRKD2 in monocytes drives increased risk of both PSC and T1DM.

Previous attempts to fine-map this locus in T1DM resulted in a credible set of 57 variants, including rs313839 among the top 10 variants[24]. Fine-mapping of this locus in the PSC GWAS data identified fourteen credible causal variants, the most probable of which was rs313839 (PP 23%), followed by rs112445263 (PP 20%). We replicated these results by fine-mapping the same locus in the colocalising monocyte eQTL data, which identified eight credible causal variants, the most probable of which were also rs313839 (PP = 14%) and rs112445263 (PP = 14%), the high LD ($r^2 = 0.98$) between these two variants accounting for their equal probability of causality. We identified a second independent signal in the PRKD2 eQTL data (PP4 55%), supported by the finding that the most probable causal configuration contained two uncorrelated SNPs; rs313839 and rs314675. The credible causal variant, rs313839, lies within an intronic region that has been associated with the binding of multiple transcription factors and overlaps a promoter for *PRKD2*.

## The UBASH3A locus

Of all PSC loci, the Chr21:43855067 locus was the most extensively investigated prior to this study. This locus was already a known eQTL of *UBASH3A* from two whole-blood and one B-cell analysis[12,26,27] and a likely shared risk locus with CeD and RhA[28,29]. We confirmed, with colocalisation, that this PSC locus shares a causal variant with CeD (PP4 = 100%), as well as T1DM (PP4 = 82%) (Fig. 4). However, evidence supporting a shared (PP4 = 42%) or different causal variant (PP3 = 54%) with RhA was equivocal.

Our colocalisation analyses confirmed that this locus is an eQTL of *UBASH3A* in T-regulatory cells (PP4 100%) and naïve CD4 + T cells (PP4 99%) from healthy recruits and T-regulatory (PP4 98%), T-memory (PP4 100%), CD4 + CCR9- effector-memory (PP4 99%), CD4 + CCR9+ effector-memory (PP4 100%) and CD8 + CCR9- effector-memory T-cells (PP4 99%) from patients with UC and PSC-UC. This locus also colocalised with an eQTL of *UBASH3A* in transverse colon tissue (PP4 95%), but not sigmoid colon, a pattern of colonic involvement reminiscent of the PSC-associated IBD phenotype. Interestingly however,

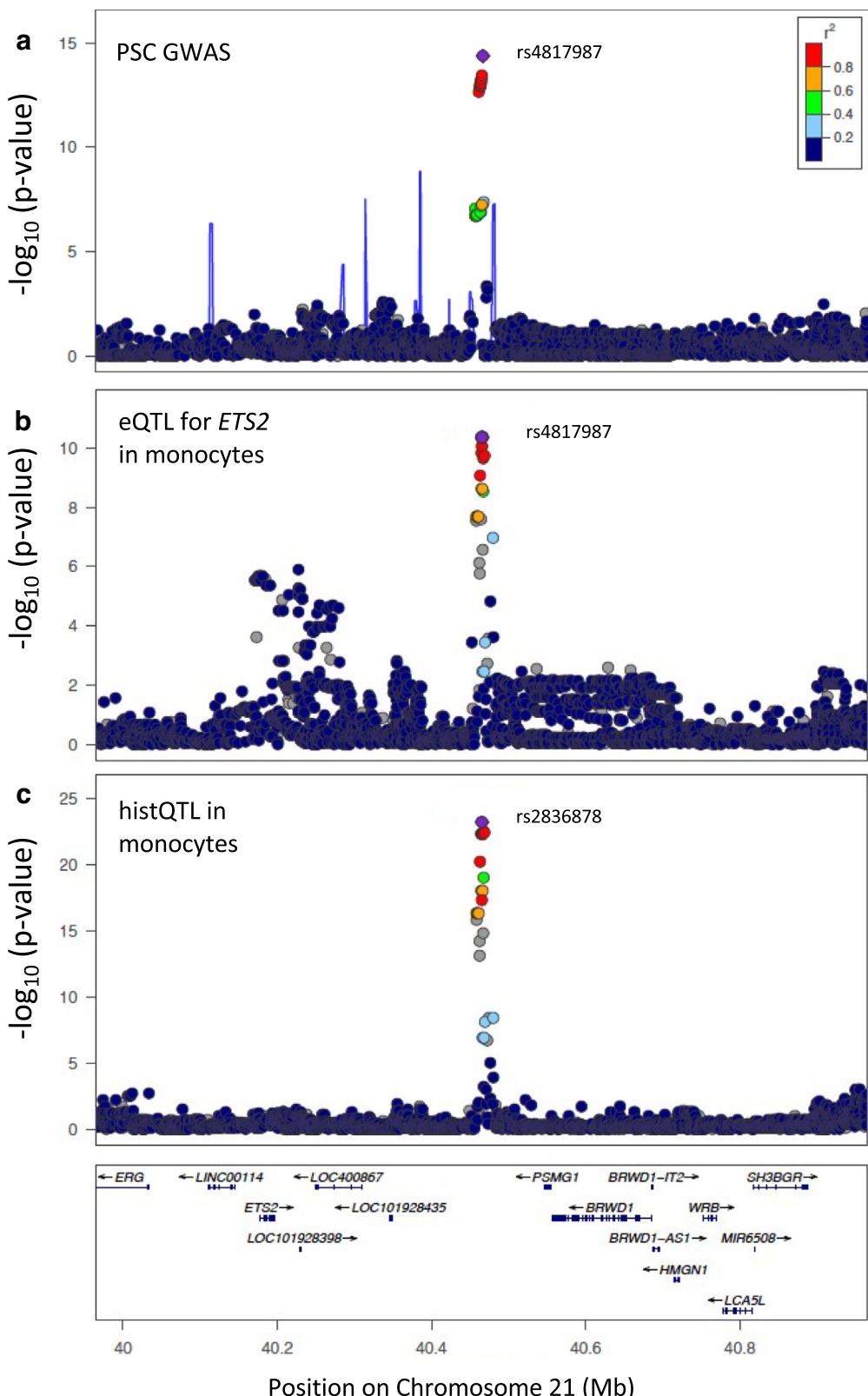

**Fig. 2 | Colocalisation results for *ETS2* Chr21:40466744.** *ETS2* Chr21:40466744 regional association plot for (**a**) PSC GWAS data ($N_{Cases}$= 4,796; $N_{Ctr}$ = 19,955), (**b**) colocalising (PP4 ≥ 0.8) eQTL data for *ETS2 in monocytes* ($N$ = 194) and (**c**) colocalising (PP4 ≥ 0.8) H3K27ac histQTL data in monocytes ($N$ = 174). The most-likely causal variant (rs4817987 PP = 0.87) is shown in purple. LD information is calculated from PSC GWAS data[9].

we found no evidence supporting a shared causal variant with UC or CD, in keeping with the results from other studies[5,7]. Our fine-mapping in the CD4 + T-cell eQTL data confirmed rs1893592 as the most probable causal variant with 99% certainty, compared to 61% in the PSC

data. In all colocalising tissue and cell-types, the PSC risk increasing rs1893592*A allele was associated with decreased expression of *UBASH3A*. This variant was previously predicted to be a splice-site variant, supported by our finding of a colocalisation with a spliceQTL in

**Table 3 | Posterior probability that each of the PSC risk loci share a causal variant with risk loci from other immune-mediated diseases**

| GWAS Signal | | | | UC | CD | PBC | T1DM | CeD | RhA | MS | SLE |
|---|---|---|---|---|---|---|---|---|---|---|---|
| Chr | Region | OR | P-value | PP4 | PP4 | PP4 | PP4 | PP4 | PP4 | PP4 | PP4 |
| 1 | MMEL1 | 1.20 | 5.12·10⁻¹³ | 0.08 | 0.00 | 0.56 | 0.01 | 0.36 | 0.45 | 0.95 | 0.02 |
| 2 | BCL2L11 | 1.29 | 2.18·10⁻¹¹ | 0.89 | 0.05 | 0.73 | 0.00 | | 0.23 | 0.00 | 0.08 |
| 2 | CD28 | 1.25 | 4.12·10⁻¹⁶ | 0.06 | 0.01 | 0.00 | 0.00 | | 0.00 | 0.00 | 0.07 |
| 3 | MST1 | 1.33 | 5.25·10⁻²⁵ | 0.85 | 0.74 | 0.01 | 0.00 | 0.08 | 0.00 | 0.00 | 0.00 |
| 3 | FOXP1 | 1.44 | 2.80·10⁻¹⁵ | 0.01 | 0.00 | 0.00 | 0.00 | 0.00 | 0.00 | 0.00 | 0.00 |
| 4 | IL2-IL21 | 1.28 | 8.25·10⁻¹⁴ | 0.02 | 0.06 | 0.56 | 0.00 | | 0.04 | 0.00 | 0.01 |
| 6 | BACH2 | 1.21 | 1.09·10⁻⁰⁹ | 0.01 | 0.07 | 0.04 | 0.18 | 0.49 | 0.87 | 0.00 | 0.02 |
| 10 | IL2RA | 1.22 | 1.44·10⁻¹⁶ | 0.05 | 0.00 | 0.00 | 0.00 | | 0.95 | | 0.00 |
| 11 | CCDC88B | 1.20 | 1.81·10⁻¹³ | 0.00 | 0.56 | 0.03 | 0.82 | 0.00 | 0.29 | 0.00 | 0.04 |
| 12 | SH2B3 | 1.18 | 3.86·10⁻¹³ | 0.89 | 0.84 | 0.94 | 1.00 | 1.00 | 0.20 | 0.00 | 0.73 |
| 16 | CLEC16A | 1.20 | 5.22·10⁻¹³ | 0.00 | 0.00 | 0.57 | 0.61 | | 0.00 | 0.00 | 0.05 |
| 18 | CD226 | 1.19 | 5.87·10⁻¹² | 0.00 | 0.03 | 0.88 | 0.76 | | 0.02 | 0.00 | 0.25 |
| 19 | PRKD2 | 1.28 | 2.12·10⁻¹² | 0.03 | 0.00 | 0.00 | 0.96 | 0.01 | | | 0.00 |
| 21 | ETS2 | 1.23 | 3.40·10⁻¹³ | 0.82 | 0.79 | 0.00 | 0.00 | 0.00 | 0.00 | | 0.00 |
| 21 | UBASH3A | 1.22 | 2.42·10⁻¹² | 0.05 | 0.00 | 0.00 | 0.82 | 1.00 | 0.42 | 0.00 | 0.00 |

OR odds ratio for lead GWAS SNP risk allele, *p*-value; for lead GWAS SNP PP H4 > 0.8, evidence for PP H3 > 0.8 underlined
PP4 for each immune-mediated disease is calculated using GWAS data from peripheral blood samples.

CD4 + T-cells (PP4 = 99%). Fine-mapping of the spliceQTL data further confirmed rs1893592 as the causal variant (PP 100%). Both observations are in agreement with a previous study that mapped eQTLs and splice QTLs across CD4 + T cells from T1DM patients[30]. The authors identified two separated molecular consequences for the PSC protective rs1893592-C allele on UBASH3A (i) a higher relative expression of the gene, but also (ii) the retention of intron 10 and 11 as a consequence of modifying a conserved nucleotide within the canonical donor site of exon 10.

### The SH2B3 locus
We fine-mapped the Chr12:11184608 PSC association signal to a single causal variant, rs3184504 (PP 99%), located in the third exon of SH2B3. Ensembl's Variant Effect Predictor (VEP) predicted the rs3184504*C > T SNP to be within the top 10% most deleterious substitutions in the human genome. We also identified colocalisations with UC, CD, PBC, T1DM and CeD, supporting a single-shared causal variant driving all six diseases.

## Discussion
We describe a PSC fine mapping study aimed to identify the most likely causal variants driving PSC risk loci and the genes they perturb, in an effort to further understand disease biology and identify drug targets. Prior to this study, twenty-three loci had been associated with PSC risk, the majority of which are in non-coding regions of the genome. Using statistical fine-mapping and colocalisation with eQTLs mapped in multiple immune-cells, including self-generated PSC T-cell eQTL maps, we have identified five genes, APO03774.1, CCDC88B, PRKD2, ETS2 and UBASH3A, perturbed by four non-coding PSC risk loci. Furthermore, we have sought to fine-map PSC risk loci and identify additional independent association signals, and the credible sets of causal variants driving each association signal. This study identifies several genes connected to the causal pathogenesis of this disease, as well as identifying four genes (ETS2, PRKD2, UBASH3A, SH2B3) involved in pathways that are targets of existing therapeutic agents or ongoing exploratory studies to develop therapeutic agents.

Our study identified colocalisation of the Chromosome 11 PSC risl locus with an eQTL for APO03774.1. APO03774.1 is a long non-coding RNA (lncRNA). LncRNA's are important regulators of both immune cell differentiation and have been implicated in the pathogenesis of multiple IMDs[31,32]. One study that mapped cis-eQTLs at 460 IMD-associated SNPs found that >10% affected the expression of a lncRNA[33]. APO03774.1, is highly expressed in PSC-relevant tissues including colon, small intestine and whole blood[34] and amongst immune cells, is most highly expressed in T-cells and NK cells, with only low expression in monocytes[22]. Interestingly, it has been previously demonstrated that expression of APO03774.1 is also associated with MS, where the lead GWAS SNP for the MS risk locus (rs694739 at Chr11:64097233) decreased the expression of APO03774.1 in PBMCs[33]. Whilst this region has not been fine-mapped in MS or any other diseases, the MS lead SNP, rs694739, which was included in our PSC data, is in high LD (r² = 0.74) with our fine-mapped SNP for this locus in PSC (rs663743 at Chr11:6410775, single variant credible set). We also found that this PSC locus colocalised with an eQTL for a second gene, CCDC88B (PP4 0.85) in monocytes, a finding also replicated in an MS study[33]. CCDC88B acts as a positive regulator of T-cell maturation and inflammatory function and loss of CCDC88B has been shown to protect against DSS-induced colitis in CCDC88B-deficient mice[35]. Recent evidence suggests that lncRNA APO03774.1 loops to a ~15 kb away regulatory element overlapping rs663743 and the promoter region for CCDC88B, which may explain the interaction between this variant and the expression of both genes[36].

Our study shows that the Chromosome 19 PSC risk locus is an eQTL of PRKD2, with the PSC risk allele resulting in reduced PRKD2 expression. PRKD2 is highly expressed in PSC-relevant tissues including whole blood, liver and colon[16] and plays a role in monocyte migration and adhesion[37]. Furthermore, Prkd2 has been identified as an inhibitory regulator of cholangiocarcinoma, a malignancy highly associated with PSC[38]. Thus, genetic variants that reduce PRKD2 expression may impair monocyte migration, subsequent tissue repair and regeneration, contributing to the PSC phenotype and subsequent neoplasia.

It has been recently shown that Prkd2 has an important role in controlling transition from naïve CD4 + T cells to T-follicular helper (TFH) cells in response to antigen or vaccine stimulus[39]. This is achieved by the direct binding and phosphorylation of Bcl6 by Prkd2, constraining Bcl6 to the cytoplasm, thereby limiting TFH development. PRKD2 loss of function mutation results in reduced expression

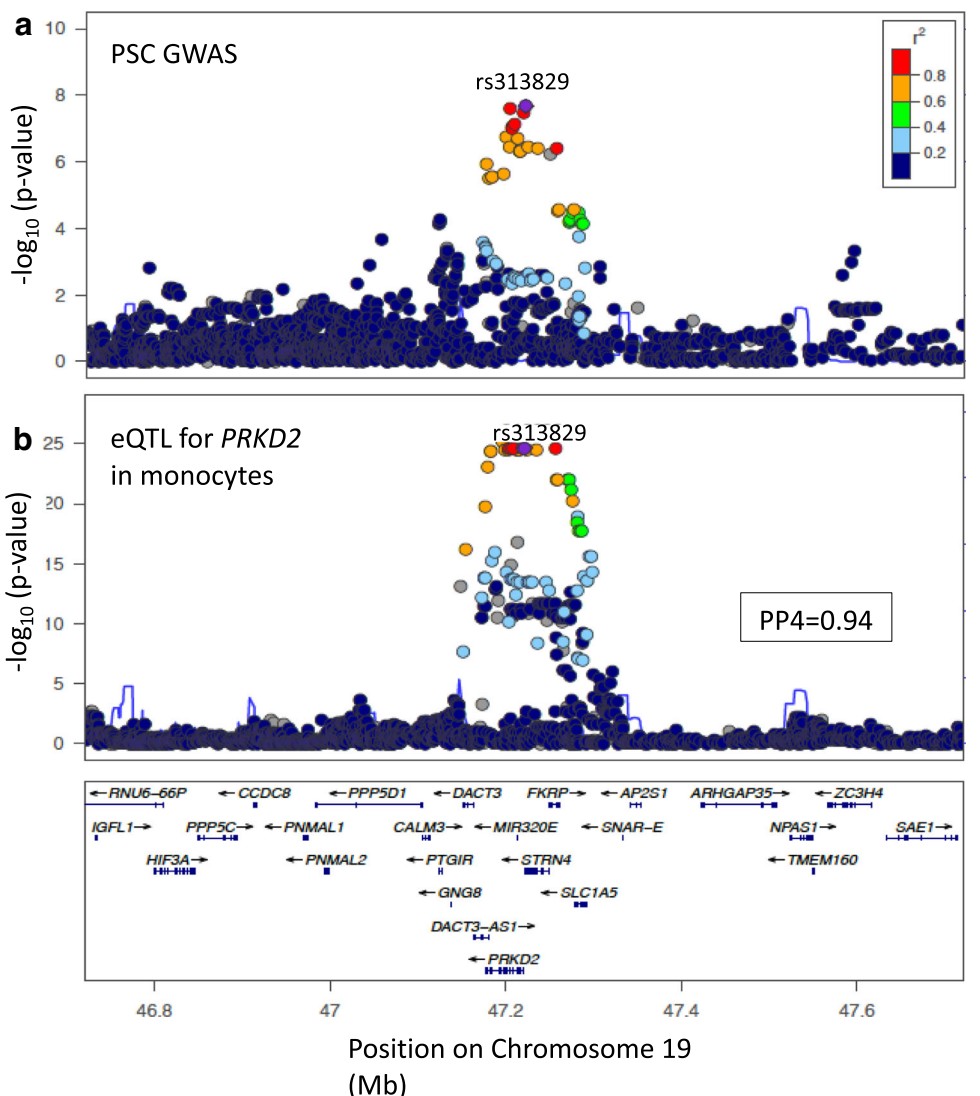

**Fig. 3 | Colocalisation results for *PRKD2* Chr19:47205707.** *PRKD2* Chr19:47205707 regional association plots for (**a**) PSC GWAS data ($N_{Cases}$ = 4796; $N_{Ctr}$ = 19,955) and (**b**) colocalising (PP4 ≥ 0.8)) eQTL data for *PRKD2* in monocytes ($N$ = 194). The most-likely causal variant (rs313829, PP = 0.94) is shown in purple. LD information is calculated from PSC GWAS data[9].

of the Prkd2 protein in mice, allowing unrestricted Bcl6 nuclear translocation in Prkd2$^{(-/-)}$ CD4 + T cells. This results in excessive cell-autonomous TFH development and B-cell activation in Prkd2$^{(-/-)}$ spleens and polyclonal hypergammaglobulinemia of IgE, IgG1 and IgA isotypes. This is particularly interesting given that TFH imbalance can contribute to IMD and IgE is often raised in the presence of IMD. Certainly, *PRKD2* has an important regulatory role in TFH development and further work examining the effects of increasing the kinase activity of Prkd2 in CD4 + T cells is warranted, not only for PSC, but also for T1DM for which this is a shared risk locus.

We demonstrate that the Chr21:40466744 locus, fine-mapped to rs2836883 is an eQTL for *ETS2*, with the PSC risk increasing allele resulting in increased expression of *ETS2* in monocytes and IL-4 stimulated macrophages. This locus is located 5′ upstream of *PSMG1*, which is the commonly quoted candidate gene for this region, based upon its genomic locality and a study of paediatric IBD colonic tissue in which *PSMG1* expression was increased compared to healthy colon[40]. However, our analysis suggests that *ETS2* is the PSC effector gene in this region and that the PSC risk increasing allele is associated with increased expression of *ETS2*. Our finding of colocalisation with an eQTL for *ETS2* in IL-4 stimulated macrophages is particularly

interesting as this is a stimulus that mimics the allergic/autoimmune response. The vast majority of iPSC-derived macrophage eQTLs are shared across the different stimulation states[41]. However, we found no evidence for colocalisation (PP4 < 0.5) in 19 of the other stimulation states. The PSC *ETS2* locus also colocalised with an association signal for neutrophil counts, where the PSC risk increasing allele was associated with a reduction in neutrophil count. This is interesting given the role of *ETS2* in inducing expression of pro-inflammatory cytokines TNFa, IL-23, IL-12, IL-1b, and chemokines, CCL2/MCP-1 and CCL3/MIP-1[41,42] in macrophages. Indeed, in mice with severe macrophage-induced pneumonitis, the prevention of Ets-2 phosphorylation results in decreased tissue macrophage infiltration[43]. Our analysis identified that this PSC locus colocalised with risk loci for both UC and CD (PP4 84% and 80% respectively), but not with any other IMDs. Thus, increased expression of *ETS2*, which has an important role in the persistent inflammatory response could contribute to driving the specific aberrant inflammatory response observed in PSC and IBD.

*ETS2* also has a role in carcinogenesis and is up-regulated in a number of cancers, notably including colorectal adenocarcinoma and hepatocellular carcinoma[44,45]. It has an important role in the Ras/Raf/MEK/ERK cascade, where *ETS2* activates the BCL-2 promoter, one of

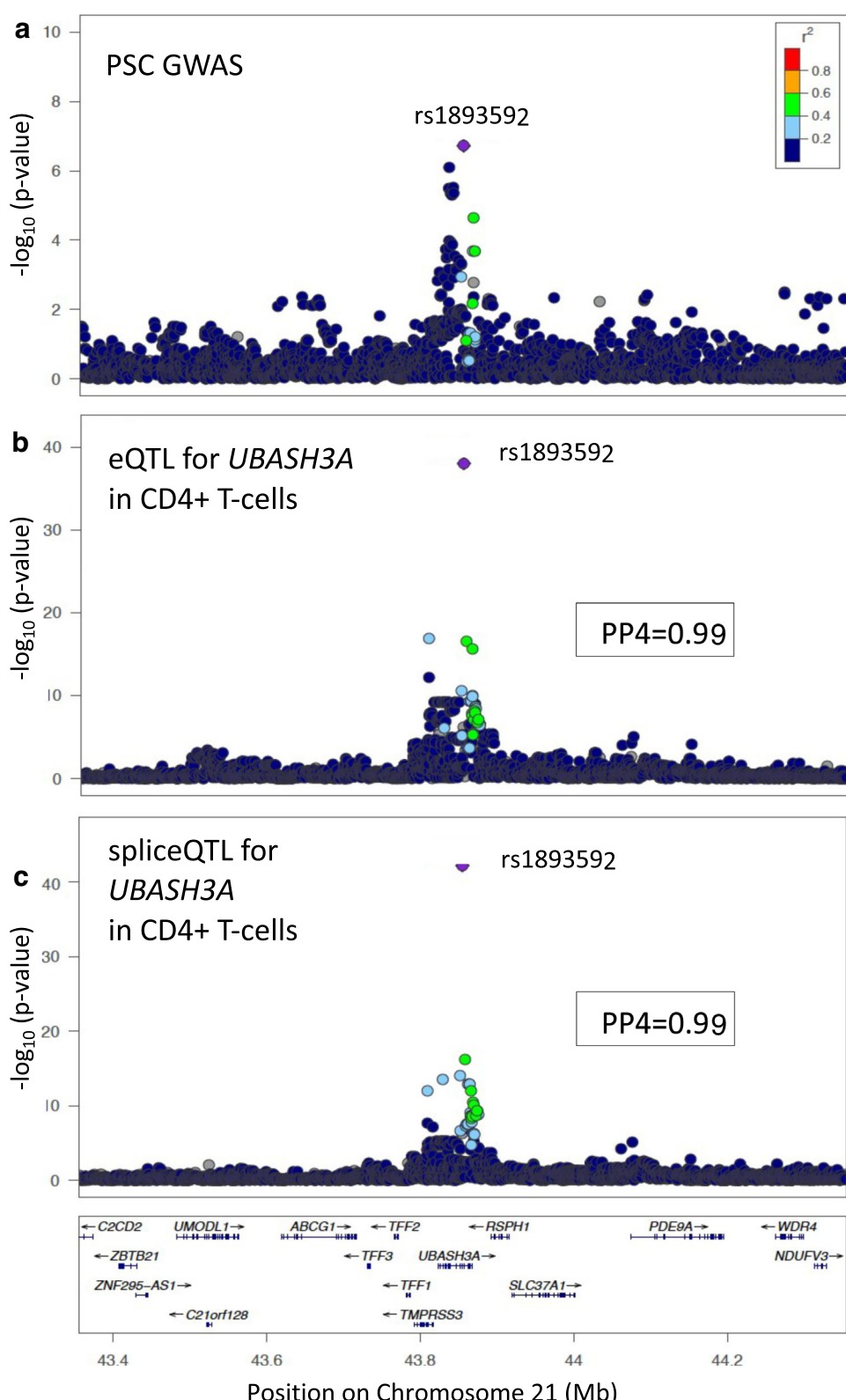

**Fig. 4 | Colocalisation results for *UBASH3A* Chr21:43855067.** *UBASH3A* Chr21:43855067 regional association plots for (**a**) PSC GWAS data ($N_{Cases} = 4796$; $N_{Ctr} = 19,955$) and (**b**) colocalising (PP4 ≥ 0.8) eQTL data for *UBASH3A* in CD4 + T- cells ($N = 171$) and (**c**) colocalising (PP4 ≥ 0.8) spliceQTL data for UBASH3A in CD4 + T-cells ($N = 171$). The most likely causal variant (rs1893592 PP4 = 0.99), is shown in purple. LD information is calculated from PSC GWAS data[9].

various apoptosis regulating factors that are phosphorylated by the Ras/Raf/MEK/ERK cascade, that subsequently inhibits cellular apoptosis[46]. For this reason, there has been recent interest in ETS2 inhibitors as a potential means of interrupting the Ras/Raf/MEK/ERK pathway and thus a potential anti-cancer therapy[47]. In PSC, *ETS2* may

contribute to several aspects of disease pathogenesis, including the induction of pro-inflammatory cytokine release from macrophages, in addition to IL-2 regulation in the transition of naive Th to Th0 cells upon antigenic stimulation. Furthermore, the role of ETS2 in the development of inflammation-induced dysplasia is yet to be explored.

Whilst work on ETS2 inhibitors is in its very early stages, our study supports further research into the mechanisms of the ETS2 pathway and its inhibition in PSC pathogenesis.

Our study confirms that the Chromosome 21 PSC risk locus is an eQTL of *UBASH3A*. We demonstrate that the rs1893592*A allele is causative for increasing PSC risk by reducing *UBASH3A* expression across almost all T-cell sub-types tested in this study, but not the wide variety of other immune cells we analysed. One criticism of performing colocalisation analyses across multiple cell types is the finding of multiple eGenes within each locus. However, consistency of both gene and cell-type, increases our confidence that we have identified the true gene or pathway affected by a genetic variant associated with disease susceptibility. Whilst there are no existing drugs targeting *UBASH3A*, this gene has an important role in the attenuation of the NF-κB/I-Kκb pathway. Proteosome inhibitors (PIs) are an existing group of drugs that target the NF-κB/I-Kκb pathway, release other pro-inflammatory cytokines and induce apoptosis of activated immune cells. PIs are currently used for the treatment of myeloma and graft-versus-host disease. Circulating proteosomes have been found in the serum of patients with several IMDs including SLE, RhA, systemic sclerosis and AIH[48] and are associated with disease progression[49]. It has been hypothesised that raised levels of circulating proteasomes function as auto-antigens[50], with anti-proteasome autoantibodies detected in the serum of patients with RhA, SLE and MS[51,52]. The immunosuppressive properties of PIs suppress the activation, proliferation, survival and immune functions of T-helper (Th) cells[53]. In RhA patients, bortezomib, inhibits the release of NF-kB-inducible cytokines by activated T-cells[50]. Thus, further research into the mechanistic effects of *UBASH3A* on the NF-κB/I-Kκb pathway in patients with PSC is warranted.

Our study fine-mapped the *SH2B3* region to a single missense variant and identified, through colocalisation, that this same variant Is associated with risk of five other IMDs. SH2B3 is a negative regulator of T-cell activation, TNF production, and Janus kinase (JAK)−2 and −3 signalling in the JAK2/STAT3 pathway. Interestingly, this is a pathway that is already the subject of successful therapeutic target in IBD and RhA[54,55] with ongoing trials in other IMDs. Our findings suggest that further investigation into the involvement of *SH2B3* and the JAKS/STAT3 pathway in PSC pathogenesis is warranted.

Our study has identified a number of causal variants, often affecting expression of target genes which contribute to complex interactions that comprise multiple immune and inflammatory pathways. Our findings require further mechanistic investigation to elucidate the exact role(s) these genes play in shaping the aberrant biology underlying PSC, and their future potential as therapeutic targets. In addition to experimental validation, future work to validate our findings could include examining the expression levels of our candidate effector genes in the disease relevant cell types or tissues in healthy individuals compared to PSC patients. Our study also demonstrates some limitations regarding the tissues in which we mapped eQTLs in PSC-UC patients. We chose peripheral blood T-cells based upon tissue accessibility, the strong HLA association with PSC and evidence for the pathogenic role of circulating peripheral blood CCR9 + T-cells in PSC. Recent evidence from a study of liver explant tissue has suggested that aberrant hepatic recruitment of gut-derived T-cells may be a shared pathogenesis with other chronic liver diseases and not unique to PSC[56]. However, this study was of patients with an end-stage rather than active disease phenotype undergoing transplantation. Furthermore, the very small numbers of patients included (NASH ($n = 3$), ALD ($n = 7$), and PSC-IBD ($n = 7$)), highlights the difficulty in obtaining liver tissue, especially from patients with PSC, where biopsy is not required for diagnosis. Despite this our study analyses were not limited to our PSC T-cell eQTL maps but included an extensive collation of QTL data covering five gastrointestinal whole-tissue types (including liver and colon), eleven immune cell-types and five different molecular QTL types. Thus, our study was widely inclusive of all tissues relevant to PSC. Another caveat of our study is that by prioritising association signals with variants reaching a large PP, we are likely to miss the small fraction of signals (-18% of cis-eQTL effects[57]) where the causal effect is driven by variants in very-high LD.

Analysis of the data generated from our PSC eQTL maps also identified causal genes in other IMDs. Colocalisation of our PSC T-cell eQTLs with IBD, rheumatoid arthritis (RhA) and Type 1 diabetes mellitus (T1DM) risk loci, identified ten IMD risk loci that colocalised with eQTLs for one or more genes. Notably, the Chr1 rs3180018 UC risk locus colocalised with an eQTL for *GBAP1* in four T-cell subtypes. Interestingly this differs from the previous candidate genes for this UC locus, *SCAMP3* and *MUC1*. However, a role for *GBAP1* in IBD is supported by the fact this same variant has been shown to increase expression of *GBAP1* in a peripheral blood eQTL study of patients with CD, resistant to anti-TNF treatment[58]. Furthermore, the UC Chr7:128573967 risk locus colocalised with an eQTL for *IRF5* in T-memory cells. *IRF5* is a transcription factor with a role in activation of pro-inflammatory cytokines IL-6, IL-12 and TNF-α[59,60], and although there are no existing drugs targeting this gene, it is widely considered to be a promising future target[58].

Common complex diseases such as IBD, RhA and T1DM have benefitted immensely from the genetics revolution. For these diseases, GWAS sample sizes now reaching the tens to hundreds of thousands have led to the discovery of increasingly large numbers of genetic risk loci. These diseases stand to benefit further from the creation of large Biobanks[61], where GWAS can be conducted at an unprecedented scale. However, for a rare disease such as PSC with a prevalence of just 1 in 10,000, the numbers of cases included within these cohorts will be too small to benefit PSC research, especially given the selection bias of many biobanks towards healthy individuals. In this study we have leveraged information from shared loci with other diseases, and increased power to fine-map molecular traits, to overcome some of the issues around small sample sizes in PSC. We have also generated a context-specific QTLs dataset, key to identifying the effector gene(s) for these loci. Continued efforts from national and international PSC consortia to create a large biobanks of genetic, genomic and clinical phenotype data from patients with PSC are needed to allow us to unfurl the complex genetic basis of this disease and its causative mechanisms to support the future development of new therapeutic targets.

## Methods

Our study complied with all relevant ethical principles in accordance to the criteria set by the Declaration of Helsinki. The study protocol and ethical approval was granted by the Norfolk and Norwich University Hospital (NNUH) Human Tissue Bank (reference number: 20122013-57 HT).

### GWAS data

We conducted fine-mapping and colocalisation analyses using genome-wide summary statistics from the best powered GWAS of PSC to date[7], with analyses restricted to the fifteen PSC risk loci outside of the HLA that reached genome wide significance in that study (4796 PSC cases 19,955 population controls, 7,891,602 SNPs tested for association). To identify PSC loci that are also associated with other immune-mediated diseases (IMDs) and/or the abundance of lymphocytes, neutrophils and monocytes, we downloaded genome-wide summary statistics for the largest published GWAS study for eight other immune-mediated diseases[28,29,46,62−65] and a GWAS of human blood cell trait variation[66] from the GWAS catalogue[67] (https://www.ebi.ac.uk/gwas/). A summary of all the utilised datasets is given in Supplementary Data 1.

## Identification of the relevant cell types

To identify immune and non-immune cell types that play a role in PSC, we used S-LDSC (v1.0.10)[68] to partition PSC heritability across a number of cell-type specific genomic annotations. These comprised a set of 53 genomic features that are known to be enriched in disease relevant variants and come packaged with S-LDSC (baselineLD_v1.2)[69,70]. We also included cell type specific annotations from two datasets, ATAC-seq data from immune cell types under resting and stimulated conditions[19] and cis candidate regulatory elements (cCRE) derived from single cell ATAC-seq data from 30 adult tissues[20]. These additional cell type specific features were incorporated into the baseline model as quantitative traits one at a time, so we could infer, and compare, the amount of heritability captured by each cell type, independently. To account for multiple testing, we estimated qvalues (R package *qvalue*) for each feature included in each independent model model (each model includes 54 features, out of these, 53 from the baseline model) and defined $q < 0.05$ as statistically significant.

## Generating PSC-specific eQTL maps

We used existing hypotheses of the causal pathogenesis of PSC, including the role of the adaptive immune system and the gut-homing (CCR9 + ) T-cell hypothesis[18], to define six PSC-specific cell-types, in which to map genetic effects on gene expression (T-regulatory, non-activated memory T-cells and activated CD4+ and CD8+ effector-memory T-cells that were positive and negative for the gut-homing ligand, CCR9). Peripheral blood was chosen due to tissue availability, because the transcriptome of end-stage cholestatic or cirrhotic explanted liver tissue is unlikely to represent the active disease transcriptome and because liver biopsy tissue is not readily available for PSC patients, for which biopsy is not a diagnostic requirement.

Seventy-six donors were recruited from the NNUH; forty-four with PSC-UC and a further thirty-two with only UC. All participants provided informed consent and were not remunerated for their participation in the study. To avoid mapping the end-stage disease transcriptome, PSC donors were all non-cirrhotic confirmed by fibroscan or biopsy (≤F3), and had a raised serum alkaline phosphatase to try and capture those with active cholestatic disease. Patients on biologic therapies were excluded, to avoid effect on the transcriptome. The inclusion of patients with only UC enabled us to increase power to detect genetic effects on gene expression that are shared across both diseases. Sex was determined by the participants assigned sex. No statistical test was used to determine sample size. We isolated peripheral blood mononuclear cells and sorted T-cell subsets using a Sony SH800 fluorescent activated cell sorter (FACS). FACS gating strategy is shown in Supplementary Fig. 6 and Supplementary Data 3. To avoid activation or degradation of cells by the sorting process, samples were processed immediately following collection at 4°C, sorted directly into chilled cell lysis buffer and immediately frozen at −80 °C. and We extracted RNA using the Qiagen RNAeasy Micro plus kit. Library preparation was performed by the Wellcome Sanger Institute Core Pipeline using NEBNext Ultra II Directional RNA kit, with a poly(A) pulldown using oligo d(T) beads and sequenced on Illumina HiSeq 4000 machines generating 75 base-pair, paired-end reads. The targeted number of reads per sample was 60 million. We aligned reads to the Genome Reference Consortium Human Build 38 and Gencode Release 29 using STAR (v2.5.3a)[71]. Read counts were assigned to genes with *FeatureCounts* (v1.5.3)[72], implemented in R (R-3.5.0). Genes with a mean expression <0.5 transcripts per million (TPMs) in all T cell subtypes were excluded from further analyses.

Differential gene expression (DGE) analyses were performed using DESeq2[73], version 1.25.0. Patient age, sex, use of drugs including 5-aminosalicylates and azathioprine, and the sequencing run were included in the model as covariates. Sample outliers were identified via PCA of the top 500 most variably expressed genes across all samples, as implemented in DESeq2, resulting in the exclusion of the CD4 +

CCR9+ gene expression data for one individual and the CD4 + CCR9- gene expression data from another individual. Two samples processed on the same day were identified as label swaps due to discordance between the inferred sex from a PCA of the expression data versus the recorded gender, and the label swap was corrected accordingly.

We extracted paired DNA for all individuals using the Qiagen DNeasy Kit. Genotyping was performed by the Wellcome Sanger Institute DNA pipelines team, using the Illumina Omni2.5-8Exome BeadChip. We performed QC on raw data using PLINK software v1.9[74], following published standards[75]. Downstream analyses used all auto-somal variants with a call rate >95% (for variants with MAF < 0.01 a more stringent call rate threshold of 98% was applied). To detect ancestry outliers we ran a principal components analysis (PCA) using PLINK v1.9's PCA function, combining LD pruned post QC genotyping data from our samples with the genotype data from 1000Genomes project (N independent variants = 62,805; excluding regions of long range LD). The PCA confirmed that all individuals were from European ancestries (Supplementary Fig. 7). Using the Wellcome Sanger Institute Imputation and Phasing Service pipeline, which runs PBWT[76], we imputed a further ~5.5 million variants from the merged UK10K + 1000 Genomes project phase 3, and the Haplotype Reference Consortium as independent reference panels. For those variants present in both panels, we retained the genotypes imputed via the Haplotype Reference Consortium data (as in Bycroft *et al*[77]). Post imputation, variants with either an imputation info <0.3 or MAF < 0.05 were removed from further analyses.

We used the MBV (Match BAM to VCF) module of QTLtools[78] to ensure genotype and gene expression data were correctly assigned to an individual, identify cross-sample contamination bias, and detect PCR amplification bias. The per-sample fraction of heterozygosity, as determined by MBV, was included as a covariate in downstream analyses as a measure to account for amplification bias.

We used QTLtools[79] to map cis-eQTLs in individual cell types. The '--normal' option was applied to rank normal transform the gene expression data, and a beta approximation permutation scheme was used to correct for the testing of multiple variants per gene[80]. We included age, sex, the fraction of heterozygosity (to account for amplification bias), three genotyping PCs (to capture intra-EUR population stratification), and a variable number of expression PCs as covariates in our eQTL model to correct for technical variation. We determined the optimal configuration of expression PCs as that which maximised the number of significant eGenes identified per T cell subtype. The final number of expression PCs ranged from 9 to 15. To correct for the thousands of genes tested per cell type, we performed an FDR correction on the set of adjusted p-values obtained by the permutation analysis, using the R package, *qvalue* and an FDR threshold of 5% to call significant eQTLs (qvalue < 0.05).

To identify eQTLs that were shared across cell-types and those that were cell-type specific we used *MashR* (multivariate adaptive shrinkage in R)[81], a method that provides improved estimates of significance and effect sizes to better consider shared effects across cell types. Following recommendations from the MashR authors, covariance matrices were defined to represent patterns of sparsity, sharing and correlation of effects between the T cell subtypes. We defined the initial 'data-driven' covariance matrices by applying PCA (mashR *cov_pca* function; N PCs = 5) to the Z scores ($N = 5487$) for the most significant eQTL associations for each significant eGene across the six PSC T cell subtypes. We then applied extreme deconvolution (mashR cov_ed) to the initial PCA-derived matrices to define the final data-driven covariance matrix estimates. To fit the model we used 200,000 random eGenes/eQTLs from our dataset as a training set, comprising both null and non-null pairs. A fitted model was then trained including the data-driven matrices together with canonical covariance matrices provided by mashR. Posterior summary statistics were computed, including effect sizes, standard errors and an estimation of the local

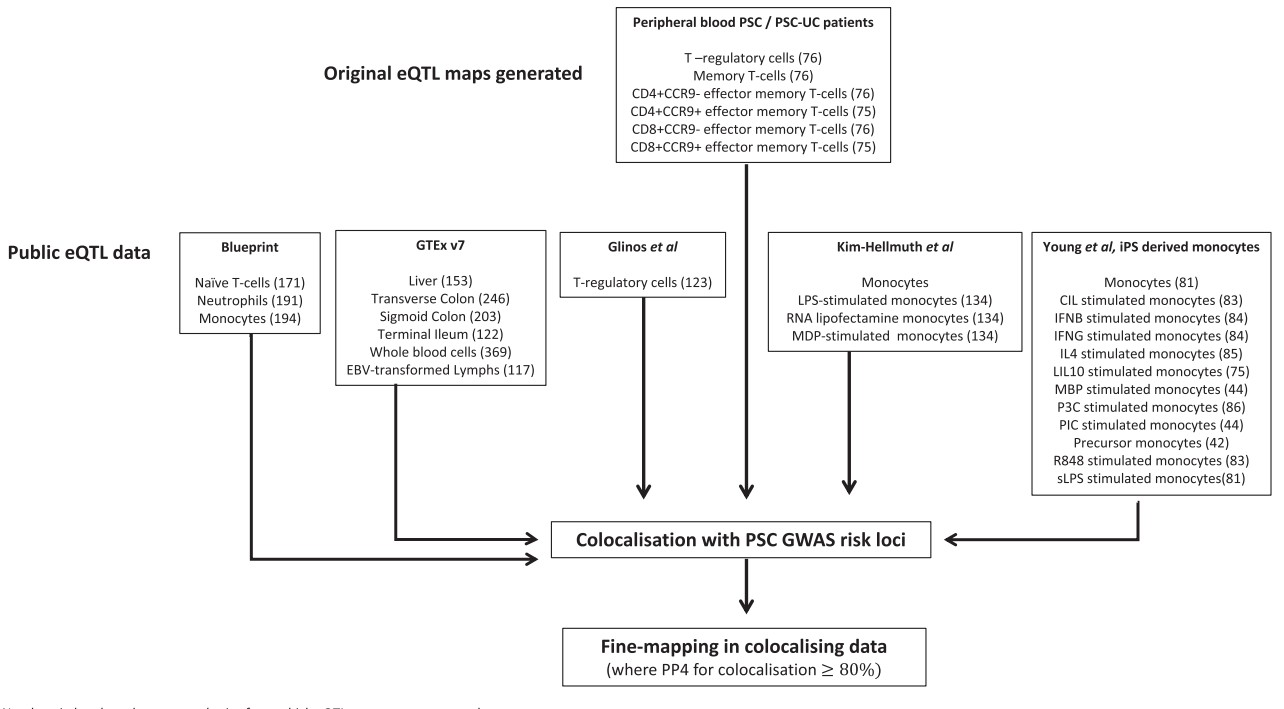

**Fig. 5 | Study flow diagram.** Study flow diagram for generation and collection of eQTL datasets.

false signed rate (lfsr). We used the lfsr to define subtype-specific effects (lfsr <0.05 in only one subtype) versus shared effects (lfsr <0.05 in more than one subtype). To estimate the proportion of significant eGenes in each of our PSC T cell subtypes that replicated in published independent eQTL datasets (including datasets accessed through the eQTL Catalogue) we estimated the pi1 statistic between cell type pairs using the R package *qvalue* (Supplementary Fig. 4)[17,82].

### Identification and ascertainment of additional molecular QTL datasets

To improve our fine-mapping, colocalisation and molecular annotation of PSC associated loci we ascertained summary-level data from published molecular QTL studies, including gene expression (eQTLs), methylation (methQTLs), histone acetylation (histQTLs) and splice site QTLs (spliceQTLs). In total, we collated 42 molecular QTL datasets, covering five gastrointestinal whole tissues, six immune-cell types and five different molecular traits including gene expression. The study flow diagram in Fig. 5 denotes all eQTL datasets used in the analysis, and supplementary data 1 denotes all eQTL and molecular QTL datasets collated for colocalisation analysis)[16,34,83–85].

### Colocalisation analysis

The fifteen non-HLA PSC loci were defined as 1 Mb regions centred on the lead variant for each locus. To interpret the direction of effect of an eQTL on gene expression in the context of the PSC risk allele, we matched QTL and GWAS reference alleles, discarding all A/T and C/G variants with a minor allele frequency (MAF) > 0.45, to minimise the chance of incorrectly matching alleles.

To test the plausibility that PSC risk loci function as eQTLs, we used the R package *Coloc* and the *coloc.abf* function to undertake per SNP Bayesian tests of colocalisation between summary statistics from PSC risk loci and each of the molecular QTL datasets (Fig. 1 and supplementary data 1). This estimates the posterior probability that a given variant is associated with neither (PP0), one (PP1/PP2) or both traits (PP3/PP4). For molecular traits with a significant association within a given PSC locus, we looked into whether the posterior

probability suggested that the two association signals were driven by two different variants (PP3) or the same causal variant (PP4). We estimated prior probabilities from our data, setting priors at $10^{-4}$ for individual trait associations and $10^{-6}$ for the probability of a SNP being associated with both traits. We determined that loci for which the posterior probability of colocalisation (PP4) exceeded 0.8 had strong evidence of a shared causal variant or 'colocalisation'.

### Fine-mapping in GWAS and functional data

To identify independent association signals, and the causal variant(s) driving each disease susceptibility association signal, we applied FINEMAP v1.3[86], a Bayesian computational fine-mapping method of GWAS summary statistics. We created the SNP correlation matrix using full original PSC genotype data[7] and the LD Store v1.1 software[87]. For each associated locus, FineMap first performs a stochastic search to ascertain the most likely number of independent effects, and then defines credible sets for each of these. To ensure our analyses satisfied the fine-mapping assumption that all potential causal variants are included within the analysis, we searched the UK10K[88] and 1000 Genomes Project[89] European population reference panels to ensure that there were no untested SNPs in high LD ($r^2 > 0.8$) with the most probable fine-mapped causal variant within a locus. Any loci failing this test were not considered to be accurately fine-mapped.

As with genetic association, power to fine-map causal variants is influenced by cohort size and phenotypic variance explained by the true causal variant (which is a function of its effect size and minor allele frequency). One challenge of studying a rare complex disease such as PSC is that amassing the GWAS samples sizes comparable to more common IMDs is not feasible. Disease-associated variants that have their effect on disease by regulating gene expression tend to exert a greater effect upon gene expression than upon complex disease risk. We, therefore, hypothesised that there may be more power to fine-map a shared causal variant using the QTL data than in the PSC data directly. Thus, for PSC loci colocalising with QTLs, we also undertook fine-mapping analyses of the colocalising QTL data.

## Reporting summary

Further information on research design is available in the Nature Portfolio Reporting Summary linked to this article.

## Data availability

Full summary statistics (nominal and permuted files) for data generated in this study are now available via Zenodo under accession code 10.5281/zenodo.11143561 [https://zenodo.org/records/11143561]. Expression data (cram files), and genotyping array data (plink format,.bed.bim.fam, as well as.idat and.gtc files) are shared via EGA (genotyping array data under Study Accession Number: EGAS00001002643; expression data under Study Accession Number: EGAS00001002642). Source data for the patient characteristics are provided with this paper in the accompanying Source Data file. Previously published data used in this study can be accessed through the GWAS catalogue: Cordell et al.[63] 2015 PBC GWAS is available at https://www.ebi.ac.uk/gwas/publications/26394269 Bradfield et al.[46] 2011 T1DM GWAS is available at https://www.ebi.ac.uk/gwas/publications/21980299 Trynka 2011 et al.[28] CeD GWAS is available at https://www.ebi.ac.uk/gwas/publications/22057235 Ocada 2014 et al. RhA GWAS is available at https://www.ebi.ac.uk/gwas/publications/24390342 Beecham 2013 et al.[65] MS GWAS is available at https://www.ebi.ac.uk/gwas/publications/24076602 Bentham 2015 et al. SLE GWAS is available at https://www.ebi.ac.uk/gwas/publications/26502338 Astle et al.[66] 2016 GWAS of lymphocyte, neutrophil and monocyte counts is available at https://www.ebi.ac.uk/gwas/publications/27863252 (monocyte counts: https://ftp.ebi.ac.uk/pub/databases/gwas/summary_statistics/GCST004001-GCST005000/GCST004625/mono_Jan2018_update/; leucocyte count: https://ftp.ebi.ac.uk/pub/databases/gwas/summary_statistics/GCST004001-GCST005000/GCST004610/wbc_Jan2018_update/; neutrophil counts: https://ftp.ebi.ac.uk/pub/databases/gwas/summary_statistics/GCST004001-GCST005000/GCST004629/neut_Jan2018_update/), De Lange et al.[62] 2017 IBD GWAS is available at https://ftp.sanger.ac.uk/pub/project/humgen/summary_statistics/human/2016-11-07/. GTEx V7 bulk tissue expression for liver, transverse colon, sigmoid colpon, terminal ileum, whole blood and RBV-transformd lymphocytes are available from the GTEx portal [https://gtexportal.org/home/downloads/adult-gtex/bulk_tissue_expression] The EGA Blueprint data for expression in naive T-cells, neutrophils and monocytes is available under accession number EGAD00001005199 Macromap data for expression of IPS-derived macrophages in multiple stimulation states is available on Zenodo [https://zenodo.org/records/7967759] Kim-Hellmuth et al.[83] 2017 expression data in monocytes is available under accession number E-MTAB-5631 Source data are provided with this paper.

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

## Acknowledgements

We are extremely grateful to all the participants who took part in this study, and the Wellcome Trust for their support in funding this study. This research was funded in part by the Wellcome Trust Grant [206194]. For the purpose of Open Access, the author has applied a CC BY public copyright licence to any Author Accepted Manuscript version arising from this submission. We would like to acknowledge the valuable support of the Human Genetics Informatics team at the Wellcome Trust Sanger Institute, UK. We would like to thank *PSC Support*, UK, who provided funding via a research grant for this study.

## Author contributions

E.C.G., L.M., N.P., T.R., S.M.R. and C.A.A. were involved in the study concept design and oversight. E.C.G., R.M., S.M.R., T.R., N.K. and A.W. were involved in design and conduct of the laboratory work. E.C.G., L.M., N.P., L.F. and B.Y.H.B. were involved in the statistical analysis. E.C.G. and C.A.A. wrote the manuscript. All authors were involved in manuscript revision and approval for submission.

## Competing interests

C.A.A. has received consultancy fees from Genomics plc and BrideBio Ltd. All other authors declare no competing interests.
