## [Peer Review File · Nature Communications]

REVIEWER COMMENTS

Reviewer #1 (Remarks to the Author):

Goode, et al. identify several genes not previously implicated to the causal pathogenesis of primary sclerosing cholangitis (PSC). First, they identify eQTLs from six PSC relevant T-cell subtypes ascertained from peripheral blood tissue of 76 PSC and Ulcerative Colitis (UC) patients. Then, they perform colocalization analysis using PSC GWAS summary statistics and publicly available eQTL summary statistics across different gastrointestinal tissues and immune cell types to identify likely causal variants and the genes they regulate, to improve biological understanding of PSC. Their discussion of four candidate associations is thoughtful and compelling. This is an important study with a unique dataset to discover eQTLs for PSC-relevant T-cell subsets. The paper is well written, and the analysis seems to have been reasonably performed. Below I identify a few elements that may benefit from more attention, and some minor comments.

Major:

I am concerned about the small sample size of the discovery dataset ($n=76$) and the impact on power. For any $MAF < 0.2$, the expected number of minor allele homozygotes is $<4\%$, so 3 or fewer individuals, and it is easy for them to drive false positive significant associations in eQTL analysis. Can the authors show the relationship between eQTL significance/effect size and MAF to quantify this phenomenon, and rule out that eQTL effects are not driven by just 1 or 2 individuals. Alternatively, focus on the heterozygote vs major homozygote comparison for less common variants.

The authors report that majority of the significant eQTLs identified from the PSC dataset are shared across all T cell types while only 5% are cell type specific. Firstly, it is not clear to me how the mashR analyses was performed to identify shared vs cell-type specific eQTLs. Could the authors elaborate on this methodology and also provide a supplementary figure showing how the effect sizes compare across T cell types or a heatmap of shared vs cell-type specific effects. Secondly, it does not answer an important question whether the shared eQTLs are shared just among T-cell subsets or with other immune cell types as well; and whether the cell-type specific ones are truly cell-type specific. Since the authors are comparing the eQTLs against multiple publicly available eQTL summary statistics, it is possible to systematically establish this result. Schmiedel et al (ref 51) would also be a good comparison dataset. Likely this work was initiated before the publication of Yazar et al (PMID: 35389779), but since that paper claims that the majority of immune eQTL are cell-type specific, they might also want to consider comparing their results to single cell eQTLs profiles.

Regarding the focused gene analyses, I also notice that SH2B3 seems to be an important gene as it shows a very high PP causal (supp table 3) with PSC GWAS risk loci, credible set size 1. It also has a PP4 > 0.8 with most of the other immune mediated diseases (table 1). Could the authors elaborate on this gene?

Table2: UBASHA3A: For this gene, it seems that eQTLs across all T-cell types are downregulating the expression for the risk allele homozygotes (negative beta of the functional trait). However, the spliceQTL is upregulating the expression in CD4 T cells, could the authors elaborate on this observation?

A possible point of discussion is the assumption that each eQTL reduces to a single SNP – multiple recent studies question this, MPRA for example providing evidence that peaks may be haplotype effects where 2 or more variants are responsible for the peak association, which need not itself be a causal variant.

In the limitations, I would also suggest providing caveats to the interpretation that each of the fine-mapped eQTL is a plausible drug candidate. While that is a strong point of the paper and it is unlikely that a drug company (or doctor) will act on the advice alone, you never know how people will over-interpret strongly argued data!

Minor:

The methods section needs more details on the steps performed –

- What were the QC steps and normalization steps performed on the cell-type specific expression data?
- Were there any batch effects and if so how were they adjusted?
- For eQTL analysis, what was the window from gene start and end used, and approximately how many SNPs per gene were used? What was the MAF threshold for inclusion of SNPs per gene? What p-value threshold was used to select significant SNP-gene pairs?
- Did the authors correct for potential confounders such as age, sex, genetic PCs, batch, for eQTL estimation?
- Please provide details on the conditional analysis and subsequently which peak was selected.
- Since the paper is based on discovery of eQTLs, please provide the full eQTL summary statistics.

- Supplementary figure 1 (mentioned in line 232) is missing.

- Table 2: GWAS risk allele for UBASH3A is missing.
- Figure 3 is not provided (it is a replica of Figure 4!)

Reviewer #2 (Remarks to the Author):

Goode et al. performed colocalization analyses using various QTL resources, including PSC-relevant T-cell subset-based eQTLs generated from PSC and UC patients, five gastrointestinal whole-tissue types, and eleven immune cell types to identify PSC-specific causal variants and provide potential drug targets at PSC risk loci.

Comments:

1. Why was the HLA region excluded from the fine-mapping analysis? Please justify the restriction.
2. While authors reported “forty-two with PSC-UC and a further thirty-four with only UC,” Figure 1 showed “Peripheral blood PSC/PSC-UC patients.” Have the authors collected the samples from PSC-UC and UC-only patients separately?
3. It seems that principal components analysis using genotype data based on 1000G population references was applied for population stratification analysis. Please provide the details – which population reference panel(s) have been used and any figures plotted by ancestry.
4. Have the authors applied any post-imputation quality controls? Please provide details about imputation analysis and any further information – genotype and gene expression data integration. The workflow chart, including each quality control and package used in each step, would be helpful.
5. Authors imputed a further ~5.5 M variants from the three reference panels. If the authors conducted them separately, how did they merge them to generate eQTL data? Also, how did they integrate them?
6. Authors estimated prior probabilities from their data. How did they calculate them?

7. Authors leveraged various public resources of eQTL data, as shown in Figure 1. It would be helpful and reproducible if the authors could add the downloadable link in Supplementary Table 2.

8. The UC and CD sample sizes from de Lange et al. (2017) include 12,160 and 12,160 in Supplementary Table 2. The sample sizes for both traits in the GWAS Catalog show 40,266 (CD) and 45,975 (UC). It seems there is a discrepancy between them. In the column of sample size, authors present only number of cases or in total?

9. In the result section of “Colocalization with molecular eQTLs,” authors wrote, “we also performed colocalization of PSC T-cell eQTLs with IBD, rheumatoid arthritis (RhA) and Type 1 diabetes mellitus (T1DM) risk loci, ... (Supplementary Table 5)”. The table title of Supplementary Table 5 presents “Results of colocalisation between T-cell eQTLs mapped in PSC-UC patients and GWAS risk loci for Ulcerative colitis (UC), Crohn’s disease (CD), Rheumatoid arthritis (RhA) and Type 1 Diabetes (T1DM).” Have the authors included IBD in the current study?

10. Authors used “colocalization” and “colocalisation” throughout the manuscript. Please choose one to make it consistent.

11. Please provide the data availability statement for the readers who want to know where and how to access data supporting the results and analysis.

12. Is this typo – “across a wide range of PSC relevant tissues”?

13. Figure 3 is incorrect. The title of Figure 3 is for PRKD2 at chromosome 19, and the figure is for ETS2 at chromosome 21.

14. The references cited in Supplementary Table 2 are incorrect. For example, while GTEx is associated with the reference #16, the reference number in the table is 18.

Reviewer #3 (Remarks to the Author):

RE: NCOMMS-23-39163

Title: "Fine-mapping and molecular 1 characterisation of primary sclerosing cholangitis genetic risk loci uncovers novel causative disease biology"

Goode et al conducted a fine mapping study aiming to identify the tissue-specific genes and variants underlying PSC GWAS loci. Primary Sclerosing Cholangitis (PSC) is a rare immune-mediated inflammatory disease. PSC confers risk of serious disease sequelae including hepatobiliary malignancy and progression to end-stage liver failure, therefore, identifying the causal genes and potential therapeutic targets would have significant public health value. The authors leveraged a number of relevant tissue(s) and molecular QTL data, and found several QTLs colocalized with PSC loci.

Major concerns:

1. Described in Method section, the disease-relevant tissues were proposed by existing hypotheses. How reliable and comprehensive are these hypotheses? Data and Methods (e.g. S-LDSC) are available to empirically identify and examine disease-relevant tissues.
2. "Causative disease biology" is an over-statement. Without experimental validation the findings should be interpreted as putative causal genes or candidate causal genes.
3. There are multiple colocalization and fine mapping tools, such as TWAS, PrediScan and SuSie. How did authors choose and justify the software applied in this study.
4. Is the FINEMAP analysis sensitive to the choice of LD matrix? If applying UK10K and 1000G LD matrix, will the results remain unchanged.
5. Beside experimental validation, another type of confirmation is to direct examine the association between PSC disease status and "putative causal gene" expression level in disease tissues.
6. In table 1, please indicate the tissue type where the PP4 was calculated.

Minor:

Please remove the edits/comments in the supplementary material file.

REVIEWER COMMENTS

Reviewer #1 (Remarks to the Author):

Goode, et al. identify several genes not previously implicated to the causal pathogenesis of primary sclerosing cholangitis (PSC). First, they identify eQTLs from six PSC relevant T-cell subtypes ascertained from peripheral blood tissue of 76 PSC and Ulcerative Colitis (UC) patients. Then, they perform colocalization analysis using PSC GWAS summary statistics and publicly available eQTL summary statistics across different gastrointestinal tissues and immune cell types to identify likely causal variants and the genes they regulate, to improve biological understanding of PSC. Their discussion of four candidate associations is thoughtful and compelling. This is an important study with a unique dataset to discover eQTLs for PSC-relevant T-cell subsets. The paper is well written, and the analysis seems to have been reasonably performed. Below I identify a few elements that may benefit from more attention, and some minor comments.

Major:

I am concerned about the small sample size of the discovery dataset ($n=76$) and the impact on power. For any $MAF < 0.2$, the expected number of minor allele homozygotes is $<4\%$, so 3 or fewer individuals, and it is easy for them to drive false positive significant associations in eQTL analysis. Can the authors show the relationship between eQTL significance/effect size and MAF to quantify this phenomenon, and rule out that eQTL effects are not driven by just 1 or 2 individuals. Alternatively, focus on the heterozygote vs major homozygote comparison for less common variants.

Many thanks for your comment. We excluded variants with a MAF threshold < 0.05 (post imputation) to help mitigate false positives due to the modest sample size. The top panel of the figure below shows the relationship between eQTL significance ($-\log_{10}$ of the corrected P-value) and MAF for the most significant variant per significant eQTL, whilst the middle panel shows the relationship between the eQTL effect size (absolute beta) and MAF across those same variants. On average, and as expected, effect sizes decay with increasing MAF because we have better power to detect smaller effects at more common variants. Reassuringly, the top panel figure demonstrates that variants of low MAF are not enriched with small p-values. Furthermore, fewer than 7% of the lead variants at significant eQTLs have a $MAF < 10\%$. Taken together, these analyses demonstrate that our results and conclusions are not negatively impacted by false-positive associations at lower frequency variants.

The authors report that majority of the significant eQTLs identified from the PSC dataset are shared across all T cell types while only 5% are cell type specific. Firstly, it is not clear to me how the mashR analyses was performed to identify shared vs cell-type specific eQTLs. Could the authors elaborate on this methodology and also provide a supplementary figure showing how the effect sizes compare across T cell types or a heatmap of shared vs cell-type specific effects. Secondly, it does not answer an important question whether the shared eQTLs are shared just among T-cell subsets or with other immune cell types as well; and whether the cell-type specific ones are truly cell-type specific. Since the authors are comparing the eQTLs against multiple publicly available eQTL summary statistics, it is possible to systematically establish this result. Schmiedel et al (ref 51) would also be a good comparison dataset. Likely this work was initiated before the publication of Yazar et al (PMID: 35389779), but since that paper claims that the majority of immune eQTL are cell-type specific, they might also want to consider comparing their results to single cell eQTLs profiles.

Many thanks for your comment and we agree that this section should be explained more thoroughly. We have expanded the paragraph in the Methods' section, describing in more detail the rationale for using MashR to identify shared eQTLs, and the steps we followed.

Methods section:

'To identify eQTLs that were shared across cell-types and those that were cell-type specific we used MashR (multivariate adaptive shrinkage in R)⁴², a method that provides improved estimates of significance and effect sizes to better consider shared effects across cell types. Following recommendations from the MashR authors, covariance matrices were defined to represent patterns of sparsity, sharing and correlation of effects between the T cell subtypes. We defined the initial 'data-driven' covariance matrices by applying PCA (mashR cov_pca function; N PCs = 5) to the Z scores (N = 5,487) for the most significant eQTL associations for each significant eGene across the six PSC T cell subtypes. We then applied extreme deconvolution (mashR cov_ed) to the initial PCA-derived matrices to define the final data-driven covariance matrix estimates. To fit the model we used 200,000 random eGenes/eQTLs from our dataset as a training set, comprising both null and non-null pairs. A fitted model was then trained including the data-driven matrices together with canonical covariance matrices provided by mashR. Posterior summary statistics were computed, including effect sizes, standard errors and an estimation of the local false signed rate (lfsr). We used the lfsr to define subtype-specific effects (lfsr <0.05 in only one subtype) versus shared effects (lfsr <0.05 in more than one subtype).'

We have also added Supplementary Figure 5, which shows a heatmap of the z-scores of significant eGenes as identified by mashR (lfsr <0.05 in at least one T cell subtype; N = 10459), grouped into those our MashR analysis classified as shared across all cell types (top panel, N = 9,176), across two or more cell types, but not in all cell types (middle panel, N = 794), and those that were only significant in one cell type (bottom panel, N = 489).

Supplementary Figure 5: Posterior z-scores of significant eGenes as identified by mashR (l_{fdr} < 0.05 in at least one PSC T cell subtype; N = 10,459), grouped by (i) significant eGenes shared across all cell types (top panel, N = 9,176); (ii), significant eGenes in two or more cell types, but not in all cell types all (middle panel, N = 794); , and (iii) those condition specific eGenes, only significant in one cell type (bottom panel, N = 489).’

Unfortunately, comparing eQTL effect sizes between different studies is not feasible due to gene expression data being normalised within studies. However, one can use the pi1 statistic [Storey J and Tibshirani R, *Proc Natl Acad Sci USA*. 2003 Vol. 100 Issue 16 Pages 9440-5] to estimate the proportion of significant eGenes shared between a pair of cell types, or tissues (as used in several studies undertaken by the GTEx consortium - for example see [GTEx Consortium. *The Genotype-Tissue Expression (GTEx) pilot analysis: multitissue gene regulation in humans. Science*. 2015 May 8;348(6235):648-60]. We used this method to identify eQTLs shared between our current study and several eQTL datasets both with and without immune cell types profiled. A description of these new analyses and results (including a new Supplementary Figure 2) is now included in the manuscript.

Supplementary Figure 2: Proportion of significant eGenes in each PSC T cell subtypes (eQTL qvalue < 0.05) replicated in external datasets (pi1)

Methods section addition:

‘To estimate the proportion of significant eGenes in each of our PSC T cell subtypes that replicated in published independent eQTL datasets (including datasets accessed through the eQTL Catalog) we estimated the pi1 statistic between cell type pairs using the R package qvalue (Supplementary Figure 2)^{17, 43}.’

Results section:

‘On average, 61-69% of the significant eGenes identified in PSC T cells replicate in GTEx tissues (estimated as the average pi1 per PSC T cell population across all GTEx tissues). The GTEx tissue with the highest replication rates was whole blood (ranging from an 88% replication rate of PSC treg eGenes to a 93% replication rate of PSC CD4posCCR9pos eGenes) (Supplementary Figure 5). The average replication rates in eQTL datasets composed of immune cell types were, as expected, larger. For example, the average replication rate per PSC T cell type in Schmiedel et al⁵² (which includes 15 immune cell types under resting and stimulated conditions) ranged between 75-82%; with the larger replication rates on this study observed with Th17 cell and Treg naive. In the blueprint data⁵³ (which includes monocytes, neutrophils and CD4+ T cells) average replication rates ranged between 90-92%, with the larger replication rates observed in CD4+ T cells.’

Regarding the focused gene analyses, I also notice that SH2B3 seems to be an important gene as it shows a very high PP causal (supp table 3) with PSC GWAS risk loci, credible set size 1. It also has a PP4 > 0.8 with most of the other immune mediated diseases (table 1). Could the authors elaborate on this gene?

Thank you for this comment regarding SH2B3, which is indeed an important gene in PSC. Due to word count restrictions we removed sections discussing this gene from the manuscript during editing, but have now reinstated these sections.

Results section:

*'We fine-mapped the Chr12:11184608 PSC association signal to a single causal variant, rs3184504 (PP 99%), located in the third exon of SH2B3. Ensembl's Variant Effect Predictor (VEP) predicted the rs3184504*C>T SNP to be within the top 10% most deleterious substitutions in the human genome. We also identified colocalisations with UC, CD, PBC, T1DM and CeD, supporting a single-shared causal variant driving all six diseases.'*

Discussion section:

'Our study fine-mapped the SH2B3 region to a single missense variant and identified, through colocalisation, that this same variant is associated with risk of five other IMDs. SH2B3 is a negative regulator of T-cell activation, TNF production, and Janus kinase (JAK)-2 and -3 signalling in the JAK2/STAT3 pathway. Interestingly, this is a pathway that is already the subject of successful therapeutic target in IBD and RHA^{78, 79} with ongoing trials in other IMDs. Our findings suggest that further investigation into the involvement of SH2B3 and the JAKS/STAT3 pathway in PSC pathogenesis is warranted.'

Table 2: UBASH3A: For this gene, it seems that eQTLs across all T-cell types are downregulating the expression for the risk allele homozygotes (negative beta of the functional trait). However, the spliceQTL is upregulating the expression in CD4 T cells, could the authors elaborate on this observation?

Many thanks for your comment. We do observe relatively greater expression of UBASH3A associated with the protective C allele (though please note that in the paper we always refer to gene expression relative to the PSC risk allele). We also show that the protective allele is predicted to disrupt the canonical donor splice site of exon 10 (exon naming in relation to the main transcript, ENST00000319294.6), and in line with this, we identify an alternative sQTL event associated with this allele. Our observations are in line with previous publications [Neman et al, Genome Res. 2017. 27: 1807-1815] where the authors identified two separated mechanisms for the rs1893592-C allele on UBASH3A in CD4+ T cells, (i) a higher relative expression of the gene, but also (ii) the retention of intron 10 and 11 as a consequence of losing the canonical donor site. We have stated those results more clearly in the text.

Results section:

'Both observations are in agreement with a previous study that mapped eQTLs and splice QTLs across CD4+ T cells from T1DM patients⁵⁷. The authors identified two separated molecular consequences for the PSC protective rs1893592-C allele on UBASH3A (i) a higher relative expression of the gene, but also (ii) the retention of intron 10 and 11 as a consequence of modifying a conserved nucleotide within the canonical donor site of exon 10.'

A possible point of discussion is the assumption that each eQTL reduces to a single SNP – multiple recent studies question this, MPRA for example providing evidence that peaks may be haplotype effects where 2 or more variants are responsible for the peak association, which need not itself be a causal variant.

Many thanks for your comment. We agree that recent MPRA studies have demonstrated that common variant association signals, for gene-expression traits or otherwise, may be driven by haplotype effects where two or more variants are underpinning the association signal. In such a scenario the most associated variant in a locus may not itself be causal for any of the independent effects, but rather it is the SNP that best captures (via LD tagging) the effects across the multiple independent causal variants, for example see [Wang, Q.S., Huang, H. *Semin Immunopathol* 44, 101–113, 2022]. Thankfully, modern fine-mapping approaches, including the FineMap method we used in this paper, have been designed with this scenario in mind and thus a typical and critical first step is to define the number of independent association signals within a locus. For each associated locus, FineMap first performs a stochastic search to ascertain the most likely number of independent effects, and then defines credible sets for each of these. In our data, FineMap found evidence of independent effects at several PSC loci and this is reported in our manuscript.

In the limitations, I would also suggest providing caveats to the interpretation that each of the fine-mapped eQTL is a plausible drug candidate. While that is a strong point of the paper and it is unlikely that a drug company (or doctor) will act on the advice alone, you never know how people will over-interpret strongly argued data!

Thank you for this excellent point. We have made the following amendments to more clearly highlight this important point.

- The title of the manuscript has been revised to remove the word ‘causative’: **‘Fine-mapping and molecular characterisation of primary sclerosing cholangitis genetic risk loci’**
- In the discussion section regarding *PRKD2* we have changed ‘therapeutic effects’ to ‘effects’: *‘Certainly, PRKD2 has an important regulatory role in TFH development and further work examining the effects of increasing the kinase activity of Prkd2 in CD4+ T cells is warranted, not only for PSC, but also for T1DM for which this is a shared risk locus.’*
- We changed the following phrase: *‘Whilst work on ETS2 inhibitors is in its very early stages, our study supports further research into the mechanisms of the ETS2 pathway and its inhibition in PSC pathogenesis.’*
- We have changed the following phrase: *‘The use of PIs in the treatment of other IMDs such as PSC is therefore a potential avenue for future investigation’* to *‘Thus, further research into the mechanistic effects of UBASH3A on the NF- κ B/I-Kkb pathway in patients with PSC is warranted.’*
- We have added an additional few sentences on the limitations of our findings with regards to therapeutics: *‘Our study has identified a number of causal variants, often affecting expression of target genes which contribute to complex interactions that comprise multiple immune and inflammatory pathways. Our findings require further mechanistic investigation to elucidate the exact role(s) these genes play in shaping the aberrant biology underlying PSC, and their future potential as therapeutic targets. In addition to experimental validation,*

future work to validate our findings could include examining the expression levels of our candidate effector genes in the disease relevant cell types or tissues in healthy individuals compared to PSC patients.'

- We have added further sentences to the discussion: *'Another caveat of our study is that by prioritising association signals with variants reaching a large PP, we are likely to miss the small fraction of signals (~18% of cis-eQTL effects⁸¹) where the causal effect is driven by variants in very-high LD.'*

Minor:

The methods section needs more details on the steps performed

- What were the QC steps and normalization steps performed on the cell-type specific expression data?

We have now added additional information about the QC and gene expression normalisation.

Methods section:

'We aligned reads to the Genome Reference Consortium Human Build 38 and Gencode Release 29 using STAR (v2.5.3a)³². Read counts were assigned to genes with FeatureCounts (v1.5.3)³³, implemented in R. Genes with a mean expression < 0.5 transcripts per million (TPMs) in all T cell subtypes were excluded from further analyses.'

'Differential gene expression (DGE) analyses were performed using DESeq2³⁴, version 1.25.0. Patient age, sex, use of drugs including 5-aminosalicylates and azathioprine, and the sequencing run were included in the model as covariates. Sample outliers were identified via PCA of the top 500 most variably expressed genes across all samples, as implemented in DESeq2, resulting in the exclusion of the CD4+CCR9+ gene expression data for one individual and the CD4+CCR9- gene expression data from another individual. Two samples processed on the same day were identified as label swaps due to discordance between the inferred sex from a PCA of the expression data versus the recorded gender, and the label swap was corrected accordingly.'

'We used the MBV (Match BAM to VCF) module of QTLtools³⁹ to ensure genotype and gene expression data were correctly assigned to an individual, identify cross-sample contamination bias, and detect PCR amplification bias. The per-sample fraction of heterozygosity, as determined by MBV, was included as a covariate in downstream analyses as a measure to account for amplification bias.'

'We used QTLtools⁴⁰ to map cis-eQTLs in individual cell types. The '--normal' option was applied to rank normal transform the gene expression data, and a beta approximation permutation scheme was used to correct for the testing of multiple variants per gene⁴¹. We included age, sex, the fraction of heterozygosity (to account for amplification bias), three genotyping PCs (to capture intra-EUR population stratification), and a variable number of expression PCs as covariates in our eQTL model to correct for technical variation. We determined the optimal configuration of expression PCs

as that which maximised the number of significant eGenes identified per T cell subtype. The final number of expression PCs ranged from 9 to 15. To correct for the thousands of genes tested per cell type, we performed an FDR correction on the set of adjusted p-values obtained by the permutation analysis, using the R package, qvalue and an FDR threshold of 5% to call significant eQTLs (qvalue <0.05). ‘

- Were there any batch effects and if so how were they adjusted?

We undertook PCA analysis via DESeq2 based on the gene expression levels of the 500 most variably expressed genes. We did not observe batch effects contributing to the largest variability observed in our dataset. Thus, PC1 and PC3 accounted for 52% and 7% of the observed variance, respectively, and were correlated with cell type; whereas PC2, which accounted for 8% of the observed variance, was correlated with the genetic sex of the participants. We included a variable number of PCs per cell type, in order to account for technical variation. We have now indicated more clearly in the manuscript the covariates included in this analysis.

Methods section

‘We included age, sex, the fraction of heterozygosity (to account for amplification bias), three genotyping PCs (to capture intra-EUR population stratification), and a variable number of expression PCs as covariates in our eQTL model to correct for technical variation. We determined the optimal configuration of expression PCs as that which maximised the number of significant eGenes identified per T cell subtype. The final number of expression PCs ranged from 9 to 15.‘

- For eQTL analysis, what was the window from gene start and end used, and approximately how many SNPs per gene were used? What was the MAF threshold for inclusion of SNPs per gene? What p-value threshold was used to select significant SNP-gene pairs?

We used the standard 1Mb centred on the TSS. The number of SNPs per gene ranged from 79 (ENSG00000275131) to 20695 (ENSG00000179344.16), with a mean of ~2600, and median of ~2500 SNPs per gene in cis (and per T cell subtype). We used a MAF frequency floor of 0.05, and a q-value threshold of 0.05 (FDR 5%) to define the significant SNP-eGene pairs. These thresholds are now stated more clearly in the manuscript. Thank you for your comment.

- Did the authors correct for potential confounders such as age, sex, genetic PCs, batch, for eQTL estimation?

A more detailed description of the covariates included in the model is now available in the manuscript.

Methods section:

'We included age, sex, the fraction of heterozygosity (to account for amplification bias), three genotyping PCs (to capture intra-EUR population stratification), and a variable number of expression PCs as covariates in our eQTL model to correct for technical variation.'

- Please provide details on the conditional analysis and subsequently which peak was selected.

The identification of independent signals was carried out by FineMap, which performs an initial stochastic search for the best model (thus the best combination of independent signals accounting for the effect(s) in the region) - and subsequently defines the credible causal set of variants within each independent signal. We have modified the text slightly to make this more clear. Thank you for your comment.

- Since the paper is based on discovery of eQTLs, please provide the full eQTL summary statistics.

We have now included in the text the following paragraph (we will provide the Zenodo link when we have the DOI for our paper):

'Full summary statistics (nominal and permuted files) are now available via Zenodo. Expression data (cram files), and genotyping array data (plink format, .bed .bim .fam, as well as .idat and .gtc files) are shared via EGA (genotyping array data under Study Accession Number: EGAS00001002643; expression data under Study Accession Number: EGAS00001002642).'

- Supplementary figure 1 (mentioned in line 232) is missing.

This was provided to the journal as a separate PDF file, which we have now added into the main collated file for ease of reference. This has been reordered and is now Supplementary Figure 4.

- Table 2: GWAS risk allele for UBASH3A is missing.

Thank you, this has been added.

- Figure 3 is not provided (it is a replica of Figure 4!)

Thank you, this error has been rectified, and the correct Figure 3 added.

Reviewer #2 (Remarks to the Author):

Goode et al. performed colocalization analyses using various QTL resources, including PSC-relevant T-cell subset-based eQTLs generated from PSC and UC patients, five gastrointestinal whole-tissue types, and eleven immune cell types to identify PSC-specific causal variants and provide potential drug targets at PSC risk loci.

Comments:

1. Why was the HLA region excluded from the fine-mapping analysis? Please justify the restriction.

As with most immune-mediated diseases, the HLA is an important mediator of disease risk in PSC with the greatest effect sizes found within this region. The extended linkage disequilibrium within the HLA makes the fine-mapping of this region more challenging and would require a different set of analyses compared to those presented in this paper. We would also require access to individual-level genotype array data to allow us to impute HLA alleles and then run fine-mapping across these and the imputed SNPs (see for instance <https://www.nature.com/articles/s41588-021-00935-7>). Since we do not have access to the genotyping array data across all IPSCSG samples, only to the summary statistics from their analyses, we cannot undertake HLA allele imputation. Thus, whilst this is indeed important future work, this was beyond the scope of this paper, where we sought to investigate only the non-coding loci on non-HLA regions with fine-mapping and colocalisation.

2. While authors reported “forty-two with PSC-UC and a further thirty-four with only UC,” Figure 1 showed “Peripheral blood PSC/PSC-UC patients.” Have the authors collected the samples from PSC-UC and UC-only patients separately?

Thank you for this comment. We collected the samples from the PSC/UC and UC patient groups separately. Following the differential gene expression steps outlined in the manuscript, which demonstrated no differences in the top 500 most differentially expressed genes between the PSC/UC and UC patient groups, we chose to collate the two patient groups samples to generate the eQTL maps. Thus the eQTL maps were based upon pooled data from the combined patient groups to improve power. We have clarified this point in the Results section of the manuscript under ‘PSC-specific T-cell eQTL data’.

‘Differential gene expression analyses failed to find any differences in gene expression between the PSC-UC and UC samples for each of the six T-cell subsets, a finding is not entirely unexpected given that both groups share the UC phenotype. Thus, we then undertook eQTL mapping in the PSC-UC and UC samples together, to maximise power.’

3. It seems that principal components analysis using genotype data based on 1000G population references was applied for population stratification analysis. Please provide the details – which population reference panel(s) have been used and any figures plotted by ancestry.

Thank you for the comment. We have now added a new sentence in the methods section specifying how the population stratification was assessed, and added a new supplementary figure with those

results. The PCA confirmed that all individuals were from European ancestries (Supplementary Figure 1). We have added the following to the methods section:

'To detect ancestry outliers we ran a principal components analysis (PCA) using PLINK v1.9's PCA function, combining LD pruned post QC genotyping data from our samples with the genotype data from 1000Genomes project (N independent variants = 62,805; excluding regions of long range LD). The PCA confirmed that all individuals were from European ancestries (Supplementary Figure 1).'

'Supplementary Figure 1: Principal component analysis of study samples compared to 1000 Genomes samples of known ethnicity using a pruned set of 62,805 independent variants with an $r^2 < 0.2$ and $MAF > 0.01$. Three individuals were of Southern European/Iberian ethnicity, highlighted on the Figure. All samples from individuals of Northern and Southern European ethnicity were retained for further analysis.'

4. Have the authors applied any post-imputation quality controls? Please provide details about imputation analysis and any further information – genotype and gene expression data integration. The workflow chart, including each quality control and package used in each step, would be helpful.

Yes indeed, thank you. We only included in further analysis those variants with $MAF \geq 0.05$; and imputation $INFO \geq 0.3$. We have now included these thresholds in the manuscript.

5. Authors imputed a further ~5.5 M variants from the three reference panels. If the authors conducted them separately, how did they merge them to generate eQTL data? Also, how did they integrate them?

The imputation of the genotyping array data was performed using the Sanger Imputation Server. The UK10K and 1000GP reference panels are already merged at the Server level, so imputation takes place taking into account samples from both cohorts. The imputation using HRC as a reference panel took place independently. To combine both datasets we followed an already established protocol [Bycroft, C *et al. Nature* 562, 203–209, 2018] where for those variants present in both panels UK10K+1000GP and HRC, only the data from HRC was retained. We have stated this now more clearly in the text.

6. Authors estimated prior probabilities from their data. How did they calculate them?

In line with recommendations from [Wakefield, J. *Genetic Epidemiology*, 33.1, 2009] for GWAS/eQTL analyses we set prior probabilities to 10^{-4} for individual trait associations and 10^{-6} for the probability of a SNP being associated with both QTL and PSC traits (denoted as p_{12}).

To define the p_{12} between PSC and other immune-mediated traits we first ran a colocalisation analysis between UC and PSC to evaluate the impact of different p_{12} priors on the PPH3 and PPH4 posteriors for a set of 7 known PSC loci.

Taking these results into account, together with Fortune *et al* [Fortune *et al. Nature genetics* 47.7 2015, p. 839] recommendations to use a p_{12} threshold between 10^{-5} and 10^{-6} , we decided to be stringent and thus set the p_{12} threshold to 10^{-6} , for our colocalization analyses.

7. Authors leveraged various public resources of eQTL data, as shown in Figure 1. It would be helpful and reproducible if the authors could add the downloadable link in Supplementary Table 2.

Many thanks, we have now added to supplementary table 1 the links to the resources.

8. The UC and CD sample sizes from de Lange et al. (2017) include 12,160 and 12,160 in Supplementary Table 2. The sample sizes for both traits in the GWAS Catalog show 40,266 (CD) and 45,975 (UC). It seems there is a discrepancy between them. In the column of sample size, authors present only number of cases or in total?

Many thanks for your raising this, that was indeed a typographic error including just the number of IBD Cases in the study. We have now updated the N in the table, also for the other studies to include the number of cases and controls from each of the GWAS.

9. In the result section of “Colocalization with molecular eQTLs,” authors wrote, “we also performed colocalization of PSC T-cell eQTLs with IBD, rheumatoid arthritis (RhA) and Type 1 diabetes mellitus (T1DM) risk loci, ... (Supplementary Table 5)”. The table title of Supplementary Table 5 presents “Results of colocalisation between T-cell eQTLs mapped in PSC-UC patients and GWAS risk loci for Ulcerative colitis (UC), Crohn’s disease (CD), Rheumatoid arthritis (RhA) and Type 1 Diabetes (T1DM).” Have the authors included IBD in the current study?

Thank you for highlighting this. We performed colocalisation with UC and CD rather than IBD collectively, and have amended the text as follows to accurately reflect this:

‘Using our own PSC T-cell eQTL maps, we also performed colocalisation of PSC T-cell eQTLs with UC, CD, rheumatoid arthritis (RhA) and Type 1 diabetes mellitus (T1DM) risk loci...’

10. Authors used “colocalization” and “colocalisation” throughout the manuscript. Please choose one to make it consistent.

Thank you we have amended the manuscript to “colocalisation” throughout the manuscript.

11. Please provide the data availability statement for the readers who want to know where and how to access data supporting the results and analysis.

Many thanks for your remark. Indeed we are making our data publicly accessible, thus we have now added the following statement (we will add the Zenodo link, once we have the DOI for our paper):

‘Data availability statement

Full summary statistics (nominal and permuted files) are now available via Zenodo. Expression data (cram files), and genotyping array data (plink format, .bed .bim .fam, as well as .idat and .gtc files) are shared via EGA (genotyping array data under Study Accession Number: EGAS00001002643; expression data under Study Accession Number: EGAS00001002642).’

12. Is this typo – “across a wide range of PSC relevant tissues”?

Thank you, we have corrected this typographic error.

13. Figure 3 is incorrect. The title of Figure 3 is for PRKD2 at chromosome 19, and the figure is for ETS2 at chromosome 21.

Thank you for noting this error, we have corrected this and added the correct Figure 3 for *PRKD2*.

14. The references cited in Supplementary Table 2 are incorrect. For example, while GTEX is associated with the reference #16, the reference number in the table is 18.

Thank you we have addressed this error.

Reviewer #3 (Remarks to the Author):

RE: NCOMMS-23-39163

Title: "Fine-mapping and molecular 1 characterisation of primary sclerosing cholangitis genetic risk loci uncovers novel causative disease biology"

Goode et al conducted a fine mapping study aiming to identify the tissue-specific genes and variants underlying PSC GWAS loci. Primary Sclerosing Cholangitis (PSC) is a rare immune-mediated inflammatory disease. PSC confers risk of serious disease sequelae including hepatobiliary malignancy and progression to end-stage liver failure, therefore, identifying the causal genes and potential therapeutic targets would have significant public health value. The authors leveraged a number of relevant tissue(s) and molecular QTL data, and found several QTLs colocalized with PSC loci.

Major concerns:

1. Described in Method section, the disease-relevant tissues were proposed by existing hypotheses. How reliable and comprehensive are these hypotheses? Data and Methods (e.g. S-LDSC) are available to empirically identify and examine disease-relevant tissues.

We thank the reviewer for this helpful suggestion. We have now undertaken heritability enrichment analyses to empirically identify disease-relevant tissues and cell types. The new paragraphs in the Methods and Results describing this analysis are shown below, along with a new figure.

New section in Methods:

'Identification of the relevant cell types

To identify immune and non-immune cell types that play a role in PSC, we used S-LDSC²⁷ to partition PSC heritability across a number of cell-type specific genomic annotations. These comprised a set of 53 genomic features that are known to be enriched in disease relevant variants and come packaged with S-LDSC (baselineLD_v1.2)^{28, 29}. We also included cell type specific annotations from two datasets, ATAC-seq data from immune cell types under resting and stimulated conditions³⁰ and cis candidate regulatory elements (cCRE) derived from single cell ATAC-seq data from 30 adult tissues³¹. These additional cell type specific features were incorporated into the baseline model as quantitative traits one at a time, so we could infer, and compare, the amount of heritability captured by each cell type, independently. To account for multiple testing, we estimated qvalues (R package qvalue) for each feature included in each independent model model (each model includes 54 features, out of these, 53 from the baseline model) and defined q-values <0.05 as statistically significant.'

New section in Results:

'PSC variants are significantly enriched at T-cell regulatory elements

To identify cell types likely to be playing a role in PSC pathogenesis we ran S-LDSC analysis across two large independent datasets, the immune cell atlas³⁰ (Supplementary Figure 3, top panel) and the cis-elements atlas of human tissues³¹ (Supplementary Figure 3, bottom panel). Within the cis-elements atlas, the most significant cell types were both T-cell populations (Lymphocyte 1 (CD8+) and T lymphocyte 2 (CD4+)), although none of the tests passed multiple testing correction. Within the immune cell atlas (Supplementary Figure 3 top panel), the only significantly enriched cell types were T-cells, both stimulated (CD8+; Central and Effector memory CD8+; Effector CD4+; Follicular T helper; Gamma delta; Memory and Naive T effector; Regulatory; Th1, Th2 and Th17 precursors) and unstimulated (Th17 precursors) from the immune cell atlas. These results support our decision to focus our de novo eQTL mapping on understudied T cell subsets of potential relevance to PSC.'

We have included a new supplementary figure as a result of this new set of analyses. See below.

Supplementary Figure 3: Enrichment analyses results. The y axis shows the enrichment (Prop.h2/Prop.SNPs) per cell type, whereas the y axis indicates the cell types. Results have been coloured by cell type, grouping different cell types in a broader category (ie, Th17 precursors, Memory Teff etc into T-cell). Top panel,,: open chromatin regions (ATAC-seq) from immune cell types under resting (_U) and stimulated (_S) conditions³⁰. Bottom panel: cis candidate regulatory elements (cCRE) inferred from single cell ATAC-seq from 30 adult tissues³¹. Significant results (qvalue < 0.05) are labelled with an asterisk.

2. “Causative disease biology” is an over-statement. Without experimental validation the findings should be interpreted as putative causal genes or candidate causal genes.

We agree with your comment. The title of the manuscript has been revised to remove the word ‘causative’ and change the title to **‘Fine-mapping and molecular characterisation of primary sclerosing cholangitis genetic risk loci nominates candidate effector genes’**.

We have also added the following paragraph to the discussion section;

‘Our study has identified a number of causal variants, often affecting expression of target genes which contribute to complex interactions that comprise multiple immune and inflammatory pathways. Our findings require further mechanistic investigation to elucidate the exact role(s) these genes play in shaping the aberrant biology underlying PSC, and their future potential as therapeutic targets. In addition to experimental validation, future work to validate our findings could include examining the expression levels of our candidate effector genes in the disease relevant cell types or tissues in healthy individuals compared to PSC patients.’

3. There are multiple colocalization and fine mapping tools, such as TWAS, PrediScan and SuSie. How did authors choose and justify the software applied in this study.

Many thanks for the question. Among the different fine mapping tools frequently used, there is not strong preference in the field between implementing FINEMAP or SuSie. Both show similar performances, with both outperforming other fine mapping tools (see for instance the comparative run in [Weisbrod *et al*, *Nat Genet* 52, 1355–1363]. We did not have access to the directly genotyped data, but only to the summary statistics, so we decided to use FineMap as it is easy to implement and the developers also provide one of the most efficient tools to infer LD (LDStore).

Regarding the strategy to define effector genes, because one of our goals was to define credible sets of causal variants via fine mapping, we decided to integrate those with a colocalization approach implemented in the R package coloc. One potential alternative could have been defining eligible target genes via TWAS or PrediScan, but that approach would not complement the first step (fine mapping) as well, and linking association signals to effector genes would have been less straightforward. One could argue that Mendelian randomization (MR) is a potential alternative to coloc but, because we wanted to focus our analyses on genome-wide significant loci, we decided to use coloc. While MR would allow us to explore non-significant signals [Zuber *et al*, *Am J Hum Genet*. 2022 May 5; 109(5): 767–782] for the majority of the eGenes we only have one cis-eQTL signal and thus lack the multiple independent instruments that MR relies on. This would have reduced our power to identify causal factors using MR.

Ultimately, every project must make difficult study design decisions regarding which data to generate and which methods to use to analyse that data. We believe the choices we have made are reasonable and sensible and have enabled us to generate robust results and insights. Of course, we do not rule out the possibility that if we had made different choices then we could have made additional insights. Given the number of different questions and analyses undertaken in the paper,

we are reluctant to add further to these, and will others to explore the utility of alternative approaches.

4. Is the FINEMAP analysis sensitive to the choice of LD matrix? If applying UK10K and 1000G LD matrix, will the results remain unchanged.

Thank you for this point. Indeed, FINEMAP (like all statistical fine mapping approaches) is sensitive to the choice of LD matrix. To explore the extent to which our fine mapping results are dependent on the choice of LD matrix we used several LD reference panels, including UK10K, 1000G and a panel derived from a subset of the PSC patients included in the PSC GWAS study from which the associated loci were defined. Reassuringly, the number of independent signals per locus, the credible set composition and the lead variants were unchanged.

5. Beside experimental validation, another type of confirmation is to directly examine the association between PSC disease status and “putative causal gene” expression level in disease tissues.

Thank you for this point. We have added the following paragraph to our Discussion section to highlight this important point:

‘In addition to experimental validation, future work to validate our findings could include examining the expression levels of our candidate effector genes in the disease relevant cell types or tissues in healthy individuals compared to PSC patients.’

6. In table 1, please indicate the tissue type where the PP4 was calculated.

Thank you, we have added a subscript to the table *‘PP4 for each immune-mediated disease is calculated using GWAS data from peripheral blood samples.’*

Minor:

Please remove the edits/comments in the supplementary material file.

Thank you, this has been done.

REVIEWERS' COMMENTS

Reviewer #1 (Remarks to the Author):

I am satisfied with the revisions and it has improved the manuscript. The only point I am still concerned about is the impact of small sample size on power. While the authors excluded variants with a MAF threshold < 0.05 and showed the relationship MAF, $-\log_{10}$ p value and effect size, it seems that there are a large number of eQTLs below MAF < 0.2 , in which case only a few minor allele homozygotes (≤ 3) may be driving the association. My suggestion would be to include a supplementary table after filtering out eQTLs below MAF < 0.2 and highlighting in the manuscript which eQTL-gene pairs still remain significant and relevant to PSC.

Other than that, please provide the minor allele tested and the minor allele frequency (MAF) for each eQTL in table 2, supplementary tables as well as in the full eQTL summary statistics.

Reviewer #2 (Remarks to the Author):

They have addressed all my concerns/comments.

Reviewer #3 (Remarks to the Author):

Title: "Fine-mapping and molecular 1 characterisation of primary sclerosing cholangitis genetic risk loci uncovers novel causative disease biology"

Goode et al presented a much-improved manuscript. I have one minor concern remaining:

The abstract mentioned "4.7% were specific to a single cell type". The eQTLs were identified at 5% FDR, so ~5% of the findings were false. The cell-type specific eQTLs could simply be the false findings (ie, random findings). I feel "4.7% were specific to a single cell type" might not be an important observation worth mentioning in the abstract or in the manuscript.

Response to reviewers' comments

Please see below our responses in blue.

Reviewer #1 (Remarks to the Author):

I am satisfied with the revisions and it has improved the manuscript. The only point I am still concerned about is the impact of small sample size on power. While the authors excluded variants with a MAF threshold < 0.05 and showed the relationship MAF, $-\log_{10}$ p value and effect size, it seems that there are a large number of eQTLs below MAF < 0.2 , in which case only a few minor allele homozygotes (≤ 3) may be driving the association. My suggestion would be to include a supplementary table after filtering out eQTLs below MAF < 0.2 and highlighting in the manuscript which eQTL-gene pairs still remain significant and relevant to PSC.

Other than that, please provide the minor allele tested and the minor allele frequency (MAF) for each eQTL in table 2, supplementary tables as well as in the full eQTL summary statistics.

Many thanks for your comment, we do appreciate your concerns. A MAF floor of 0.05 is frequently applied to eQTLs studies with similar sample sizes to ours (see for instance Schwartzenruber, Nat Genet. 2018 or Naranbhai, Nat Comms, 2015). Given this, we are reluctant to throw doubt over significant eQTLs we identified with MAF between 0.05 and 0.20 out of fear these signals *may* be driven solely by a few minor allele homozygotes. Instead, we decided to test the hypothesis that these signals are driven by the few minor allele homozygotes. To do this, we fitted a dominant model to test for a cis-eQTL effect at each of the lead SNPs from the set of previously identified eQTLs with a MAF between 0.05 (our MAF floor) and 0.2. This model tests for a difference in mean gene expression between the WT homozygous individuals versus both the heterozygous and ALT homozygous individuals *together*. As such, the mean gene expression of ALT homozygous individuals will have a greatly reduced effect on the outcome of the statistical test.

Reassuringly, the significance of the additive and dominant tests were highly correlated ($r^2 = 0.93$) across the set of 2,595 lead SNPs with MAF between 0.05 and 0.2 (Figure 1, left panel). Furthermore, no significant differences in effect were detected between the additive and dominant model after correction for multiple testing (min heterogeneity p-value=0.45; Figure 1, right panel). Figure 2 shows eQTL effects for 12 randomly selected eQTLs with MAF between 0.05 and 0.2, for both the additive and dominant model. This shows that many of the QTLs identified don't have any WT homozygous

individuals, and differences in gene expression between the WT homozygous individuals and heterozygous individuals are visible.

Given these reassuring results, we hope the reviewer will agree with us that our QC and stringent significant thresholds have yielded robust eQTL effects, even for SNPs with minor allele frequencies between 0.05 and 0.2.

Figure 1

Figure 2

Reviewer #2 (Remarks to the Author):

They have addressed all my concerns/comments.

Thank you.

Reviewer #3 (Remarks to the Author):

Title: "Fine-mapping and molecular 1 characterisation of primary sclerosing cholangitis genetic risk loci uncovers novel causative disease biology"

Goode et al presented a much-improved manuscript. I have one minor concern remaining:

The abstract mentioned "4.7% were specific to a single cell type". The eQTLs were identified at 5% FDR, so ~5% of the findings were false. The cell-type specific eQTLs could simply be the false findings (ie, random findings). I feel "4.7% were specific to a single cell type" might not be an important observation worth mentioning in the abstract or in the manuscript.

Many thanks for your comment. Inferences regarding cell type specificity were not based on FDR estimates. Rather, we applied mashR, a multivariate adaptive shrinkage method, to more accurately compare eQTL effects between tissues (Urbut et al., Nat Genet, 2018). Briefly, mashR has two key differences from standard approaches to FDR analysis. First, it assumes that the distribution of the actual (unobserved) effects across conditions is unimodal, with a mode at 0. Second, the method takes as its input two numbers for each test (an effect size estimate and corresponding standard error), rather than the one number usually used (p value or z score). Using these two numbers helps account for variation in measurement precision across tests. It also facilitates estimation of effects, and unlike standard FDR methods, the approach provides interval estimates (credible regions) for each effect in addition to measures of significance. To provide a bridge between interval estimates and significance measures, mashR calculates a "local false sign rate", which is the probability of getting the sign of an effect wrong. The mashR authors demonstrate it is a superior measure of significance than the local FDR because it is both more generally applicable and can be more robustly estimated (due to the underlying assumptions of the mashR model outlined above) (Stephens, Biostatistics, 2017).

We required eQTLs to have an *lfsr* threshold of less than 0.05 in a single cell type to define cell-type specific effects. We recognise that this threshold is as arbitrary as any

FDR or p-value threshold, and therefore any change to it will result in a different proportion of cell-type specific eQTLs. While it is true that the proportion of cell-type specific effects (0.047) is similar to our *lfr* threshold, we don't believe this is because these cell-type specific effects are largely false-positives. The approach we have taken to declaring cell-type specific eQTL effects is conservative. We have been clear about the approach we have taken, including the FDR threshold for defining eQTLs and the *lfr* threshold for declaring cell-type specific effects. We believe the rate of cell-type specific eQTL effects is an important quantity to estimate because it allows researchers to evaluate the value of performing eQTL mapping experiments in particular disease relevant cell-types. Targeting such cell-types was a central aim of our project, so it seems pertinent to quantify in retrospect how useful this approach was. With this in mind, we would like to keep the proportion of cell-type specific effects in the manuscript. As a compromise, we have removed the estimate from the abstract.